# Protein Set Transformer: a protein-based genome language model to power high-diversity viromics

Cody Martin [1,2], Anthony Gitter [3,4,5] & Karthik Anantharaman [1,6,7] ✉

Exponential increases in microbial and viral genomic data demand transformational advances in scalable, generalizable frameworks for their interpretation. Standard homology-based functional analyses are hindered by the rapid divergence of microbial and especially viral genomes and proteins that significantly decreases the volume of usable data. Here, we present Protein Set Transformer (PST), a protein-based genome language model that models genomes as sets of proteins without considering sparsely available functional labels. Trained on >100k viruses, PST outperforms other homology- and language model-based approaches for relating viral genomes based on shared protein content. Further, PST demonstrates protein structural and functional awareness by clustering capsid-fold-containing proteins with known capsid proteins and uniquely clustering late gene proteins within related viruses. Our data establish PST as a valuable method for diverse viral genomics, ecology, and evolutionary applications. We posit that the PST framework can be a foundation model for microbial genomics when trained on suitable data.

Viruses are the most abundant biological entity on the planet and inhabit every ecosystem. Understanding how viruses modulate microbiome dynamics and function is an active area of research that spans various scales from global biogeochemistry[1] to human health and disease[2]. However, comprehensive large-scale viral metagenomics (viromics) studies are hindered by the enormous genetic diversity of viruses, as most genomics tools rely on sequence similarity to existing reference databases. These problems are compounded by the lack of universal genes in viruses, complicating phylogenetic and comparative analyses across diverse groups of viruses. Overall, these challenges have impeded the development of viromics software that is both accurate and scalable to increasingly diverse viral datasets. Thus, there is a clear need to develop data-driven frameworks to study viruses at-scale using generalizable genomic principles instead of simple sequence homology-based methods.

Protein language models (pLMs) are promising deep learning frameworks for generalizable genomics. Trained on corpuses of millions of proteins[3–5], pLMs have learned biochemical, functional, and structural features of proteins from contextual information of amino acids patterns. Applications of pLMs to viral datasets have demonstrated increased capacity for protein function annotation[6,7] and host prediction[8]. However, these studies only focused on specific tasks without considering that pLMs could be universally beneficial for a variety of viromics tasks[9], underutilizing the full potential of foundation pLMs. An additional shortcoming of pLMs themselves is that they do not account for evolution-driven genome organization. Recent work has addressed this issue by contextualizing pLM embeddings across short genomic fragments[10] and representing the entire genome as an aggregation of the pLM embeddings[8]. However, each of these models only targets one kind of representation: the former represents proteins with added genome context, while the latter represents

[1]Department of Bacteriology, University of Wisconsin-Madison, Madison, WI, USA. [2]Microbiology Doctoral Training Program, University of Wisconsin-Madison, Madison, WI, USA. [3]Department of Biostatistics and Medical Informatics, University of Wisconsin-Madison, Madison, WI, USA. [4]Morgridge Institute for Research, Madison, WI, USA. [5]Department of Computer Sciences, University of Wisconsin-Madison, Madison, WI, USA. [6]Department of Integrative Biology, University of Wisconsin-Madison, Madison, WI, USA. [7]Department of Data Science and AI, Wadhwani School of Data Science and AI, Indian Institute of Technology Madras, Chennai, TN, India. ✉e-mail: karthik@bact.wisc.edu

genomes as a weighted sum of protein embeddings subject to a specific classification task. Further, a similar method classified plasmid proteins ontologically[11], but the supervised training prevents generalizing to related tasks that could benefit from the same contextualization approach. Thus, none of these approaches are truly generalizable to a variety of viromics tasks that require both protein- and genome-level reasoning.

Here, we present Protein Set Transformer (PST), a protein-based genome language model that uses an encoder-decoder paradigm to simultaneously produce both genome-contextualized protein embeddings and genome-level embeddings within a single end-to-end model. We developed several algorithmic advances for the PST architecture and implemented more memory-efficient data handling compared to similar methods[8,10,11]. We additionally pretrained foundation PST models on >100k high-quality dereplicated viral genomes encoding >6 M proteins using a self-supervised objective. Then, PST was evaluated on two distinct test datasets, one consisting of >150k high-quality viral genomes encoding >7 M proteins from IMG/VR v4[12] and another of >12 k viruses encoding >141 k proteins detected from soil metagenomes deposited in MGnify[13]. We demonstrate that PST better captures viral genome-genome relationships based on shared protein content and is highly sensitive to remote relationships. Further, we observe that only PST can consistently cluster functionally related proteins like late gene proteins, indicating the importance of genome context-aware training. Additionally, PST protein embeddings are associated with protein structure relationships, as demonstrated by clustering proteins with capsid folds that could not be detected by sequence similarity with annotated capsid proteins. Notably, neither the genome-contextualized PST protein embeddings nor the genome embeddings were learned with respect to any external labels, meaning that they will be useful for a wide range of applications. This flexibility of our pretrained PST means it can be used for transfer learning to other viral-centric tasks, such as viral gene and genome identification, genome quality control, genome binning, taxonomy, and host prediction, which are major components of viromics research[9]. For example, we show that PST substantially improves viral-host prediction compared to existing prediction tools. Thus, we expect that PST will be integral to future viromics studies. Further, we posit that the PST architecture can be a general-purpose model for microbial genomics when trained on microbial instead of, or in addition to, viral genomes.

## Results

### Developing the PST genome language model

PST (Fig. 1a, Supplementary Fig. 1) models genomes as sets of proteins using principles from the natural language processing and set[14] and pointset[15] transformer[16] fields. We refer to PST as a protein-based genome language model since it contextualizes protein information at genome-scale. In PST, all proteins from each genome are embedded using the well-established ESM2 pLM[3,4]. Unlike Set Transformer[14], PST concatenates small vectors onto the pLM embeddings to model both the protein genome position and coding strand. The updated protein embeddings from each genome are fed to the PST encoder, which uses multi-head attention[16] to contextualize the protein representations within each genome. These PST protein embeddings can be used for protein-level tasks like protein classification and functional annotation. In an end-to-end PST, the PST protein embeddings are passed to the PST decoder, which also uses multi-head attention to weigh the relative importance of each protein in a genome. These weights are used for a weighted average of the contextualized proteins to produce a genome representation.

A common training objective for language models that is used by a similar protein-based genome language model[10] is masked language modeling (MLM)[17], which involves predicting masked tokens (words) in sentences from the rest of the sentence as a classification task. In the

case of genome sentences composed of protein words represented as dense vectors, MLM is less intuitive and likely overcomplicates training. For example, the number of distinct proteins massively outnumbers the number of unique words in natural languages. Additionally, the representation of proteins as embeddings prevents using traditional classification-based objective functions and requires customized regression objectives to tailor these domain differences to fit MLM[10].

We instead opt to mirror relationship-guided genomics to better understand patterns of genetic diversity using the triplet loss (TL) function[15,18] (Fig. 1b, d). During self-supervised pretraining of the PST foundation model, TL uses the distance in PST embedding space as a measure of genome-genome relatedness, which are implicitly conditioned on protein-protein relatedness in PST. Briefly, TL involves the formation of genome triplets, consisting of one as an anchor, the genome most related to the anchor as a positive example, and a genome less similar than the positive genome as a negative example[15,18] (Fig. 1b, d). Positive examples are defined using the Chamfer distance in the input embedding space among genomes within a training minibatch. Chamfer distance is computed as the average minimum of protein-protein embedding distances for pairs of genomes, meaning that the positive genome has the most similar proteins to the anchor genome. Meanwhile, negative examples are sampled in the PST embedding space using a semi-hard negative sampling strategy to select negative genomes that are closest to but slightly farther than the positive genome. Since PST is trained with a self-supervised objective, there are no obvious choices of negative examples, so the chosen negative example is weighted based on the Chamfer distance between the anchor and negative genomes. This notably reduces the contribution to the loss for negative genomes that are actually similar to the anchor based on Chamfer distance (see Methods). Overall, the objective of TL is to embed the positive genome closer to the anchor than the negative within a tunable margin (Fig. 1d).

To help PST learn more generalizable representations, we used the data augmentation technique PointSwap[15] (Fig. 1c) for each genome and its most related genome defined by Chamfer distance above (Fig. 1b). Each genome pair swaps protein vectors that are most similar at a defined, tunable rate, analogous to homologous recombination. We then update the TL objective to include maximizing the similarity between the anchor genome and its corresponding augmented hybrid genome produced by PointSwap.

We additionally trained an encoder-only PST (Supplementary Fig. 1) with a MLM-style objective to compare the choice of objective for viral genome and protein representation learning. During training, proteins are masked at a tunable rate in each genome as targets. In traditional MLM, masked word predictions are scored using a classification-based objective. Here, however, protein words are represented as dense vectors in genome sentences, which does not easily allow for simple classification. We, therefore, adapted a previous approach[10] that uses a mean squared error loss function to consider the distance between the predicted protein embeddings and the masked target embeddings. Notably, this also enables a more direct comparison against a previously published genome language model that is similar to PST[10]. Further, to model real genetic diversity and the fact that different types of proteins can occur in the same genomic contexts, we augment the MLM objective by positively sampling the nearest protein embedding in ESM2 space to create positive examples. Our MLM objective, thus, considers distance between the predicted protein embedding and the masked targets as well as the distance between the predicted positive protein embedding and the masked targets. We refer to the full encoder-decoder PST trained with TL as "PST-TL" and the encoder-only PST trained with MLM as "PST-MLM". References to a generic "PST" indicate a full encoder-decoder PST (i.e., PST-TL), unless otherwise specified.

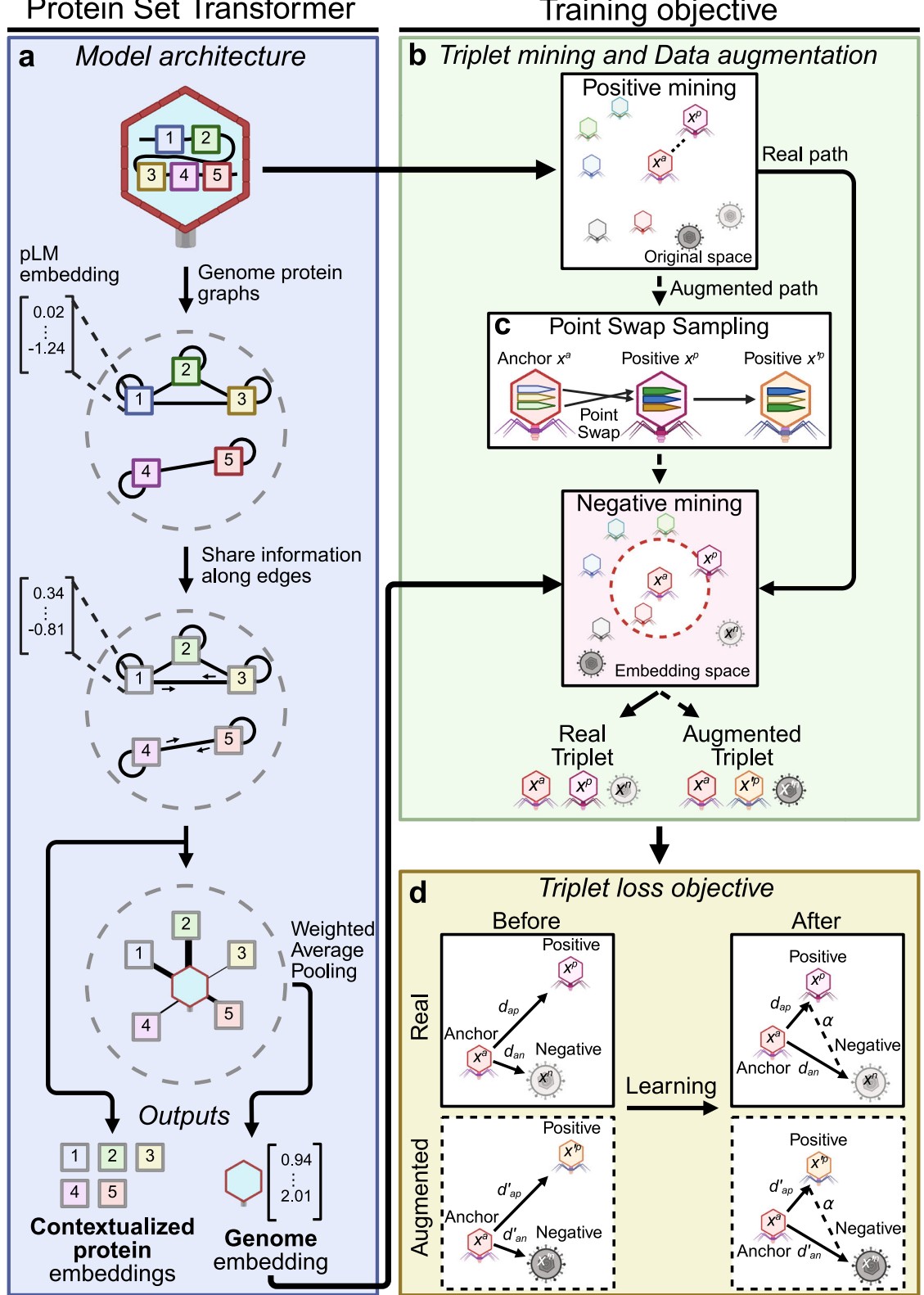

**Tuning PST using two cross validation strategies**

To train PST foundation models, we collected 103,589 viruses from 12 different publicly available sources[1,19–29] as a training dataset (Supplementary Fig. 2a, Supplementary Data 1). We evaluated PSTs on two distinct test datasets: 151,255 viruses from IMG/VR v4[12] and 12,857 viruses detected from soil metagenomes deposited in MGnify[13]. The viruses from each test set were chosen since they were distinct at the nucleotide level (<95% ANI over 85% of either genome) from the

training viruses. The MGnify test viruses were further dereplicated with the IMG/VR v4 test viruses using the same nucleotide similarity criteria as above. Dereplication at the nucleotide level was sufficient to reduce train-test genome similarity at the protein level (Supplementary Fig. 3b, d). Most viruses in each set were predicted to encode between 2–100 proteins (Supplementary Fig. 2b) and to be Duplodnaviria, Monodnaviria, or Riboviria (Supplementary Fig. 2c). Additionally, most were from environmental sources (Supplementary Fig. 2c), especially

**Fig. 1 | The Protein Set Transformer (PST) architecture and training regime.**
**a** General overview of the graph-based PST for learning genome representations from contextualized protein embeddings. Each protein is represented by an ESM2 protein embedding. PST internally represents each genome as a graph, consisting of multiple subgraphs of fully connected, locally adjacent proteins. The size of each subgraph is a tuned hyperparameter. PST uses multi-head attention both to contextualize protein embeddings within each genome and to learn per-protein weights for a weighted averaged over each genome. See Supplementary Fig. 1 for a modeling-centric view of PST. Both protein and genome representations can be used for an appropriate downstream task. **b** Triplet mining workflow that includes the data augmentation technique **c** PointSwap sampling. For each training genome, a positive genome is identified from the ESM2 embedding space defined as the minimum Chamfer distance. Then, a negative, less related, genome is chosen from the PST embedding space that is the next farther genome after the positive. We augment our training data by creating hybrid genomes that swap similar protein vectors between each genome and its positive genome. **d** Pictorial representation of the triplet loss objective function used to train PST on viral genomes. The operational objective of triplet loss is to embed each genome and its positive genome closer in embedding space than each genome and its negative genome, within a tunable distance margin.

marine systems. Among the viruses with a known or predictable host, most were bacterial viruses (Supplementary Fig. 2c).

We tuned eight different PSTs starting with small (6-layer, 8 M param) or large (30-layer, 150 M param) ESM2 protein embeddings, respectively, using the TL and MLM objectives. Four models were tuned using a variant of leave-one-group-out (LOGO) cross validation (CV), where the group is the viral taxonomic realm. In our variation, the Duplodnaviria group is always included as a CV training fold since this group composes a significant fraction of our training set (65.4%, Supplementary Fig. 2c). This CV setup notably helps choose model hyperparameters optimal for all viruses rather than just the most abundant, since the validation groups never include this substantially large viral taxonomic realm. "T" is appended to the name of these models (i.e. PST-TL-T).

The other four models were tuned using CV groups based on protein diversity, such that there was minimal overlap of shared protein content between these groups (Supplementary Fig. 4c). We considered this alternate approach because viral taxonomy may not necessarily correlate with the protein diversity and evolution found in viral genomes. For example, the amount of protein diversity within the Duplodnaviria may be far greater than other viral realms simply because it is a much larger group, meaning that these taxonomic groups are not very homogeneous. "P" is appended to the names of these models (i.e., PST-TL-P).

The best models were chosen based on the lowest validation loss averaged among all folds at the end of tuning. Using this strategy, we tuned training-specific (dropout, layer dropout, learning rate, weight decay, and batch size), model-specific (number of attention heads and encoder layers), PST-specific (chunk size), PointSwap-specific (rate), and TL-specific (distance margin, scale factor) hyperparameters (Supplementary Figs. 5,6, Supplementary Tables 1,2). We additionally tuned a masking rate for PST-MLM (Supplementary Fig. 6). For PST-TL-T_small, fewer attention heads and encoder layers led to optimal performance, while the reverse is true for PST-TL-T_large, likely reflecting the increased information capacity of larger pLM embeddings. There were no striking differences between the PST-TL-P models concerning model complexity. Increasing values of the PointSwap rate generally led to decreased loss for all PST-TL models. Notably, increasing values of the TL distance margin led to greater performance for PST-TL-T, while decreasing this resulted in somewhat better performance for PST-TL-P. For PST-MLM models, there were not many major trends.

After 20–75 hyperparameter tuning trials (Supplementary Table 3), we trained a final model for each model size, CV strategy, and objective using the best hyperparameters (Supplementary Fig. 7, Supplementary Table 2). The remaining results are based on these models that we refer to using the following format: "PST-OBJECTIVE-CV_SIZE", i.e., PST-TL-T_small indicates a full encoder-decoder PST trained with TL, tuned with the taxonomic CV strategy, and using the ESM_small protein embeddings as input. The model sizes are both 5.4 M for both small PST-TL models, 21.3–177.9 M for large PST-TL models, 23.8–93.0 M for small PST-MLM models, and 93.6–185.8 M for large PST-MLM models (Supplementary Table 4).

## PST captures viral genome-genome evolutionary relationships

To evaluate if PST learned biologically meaningful representations of viral genomes, we compared the genome embeddings produced by PST against other protein- and nucleotide-based methods with a quantitative clustering assessment on the test datasets. For protein-based methods, we performed an ablation study comparing unweighted averages of the input ESM2 embeddings and of the PST protein embeddings over each genome ("PST-CTX" methods). We additionally compared the choice of TL or MLM training objectives. For nucleotide-based methods, we used 4-mer nucleotide frequency vectors, GenSLM[30] embeddings, and HyenaDNA[31] embeddings. The latter methods were chosen for both their availability relative to the course of this study and their relevance to genome language modeling, as there have been several recently described nucleotide-based models[32,33]. GenSLM and HyenaDNA have also been referred to as genome language models, so we explicitly refer to these as nucleotide language models to distinguish them from our protein-based genome language model. GenSLM was trained to focus on codon-level words in a genome sentence. Thus, to produce GenSLM genome embeddings, we embedded each nucleotide open reading frame (ORF) and then averaged these over each genome. Meanwhile, HyenaDNA is a long-context nucleotide language model that can contextualize up to 1 Mb fragments, which is well above the size of most viral genomes. Protein-based methods and HyenaDNA appeared to better reflect the evolutionary relationships among viruses in both the PST training and IMG/VR v4 test datasets in a qualitative analysis of the genome embeddings in which there are visually distinct clusters of the four viral taxonomic realms (Fig. 2a).

To quantitatively evaluate each genome representation, a similarity-weighted k-nearest neighbor (kNN) graph was constructed from each of the genome embeddings from each test dataset with 15 neighbors and then clustered using the Leiden algorithm[34] with a similarity threshold of 0.9. This led to 20–30k clusters for the IMG/VR v4 test set (Supplementary Fig. 8a) and 1.5-2.5k clusters for the MGnify test set (Supplementary Fig. 8b). For both datasets, the average cluster size ranged from 4–8 genomes depending on the genome embedding (Supplementary Fig. 8). Notably, this led to ≥75% of all viral genomes in each dataset being clustered with at least 1 other genome (Supplementary Fig. 8).

For pairs of genomes in each genome cluster, we computed the average amino acid identity (AAI) and average structural identity (ASI; see Methods). Since AAI is based on protein sequence alignments and ASI is based on protein structure alignments, these two metrics allow us to distinguish between genomes that are closely related and remotely related. This is because AAI relies on protein sequence alignments, which are not as sensitive as structural alignments. Likewise, genomes where AAI cannot be detected but ASI can be detected are considered distantly related, since there are protein structure alignments but no protein sequence alignments.

We then plotted a score aggregating ASI with the proportion of shared protein structures against the embedding angular similarity (Supplementary Figs. 9–12) and computed the Pearson correlation coefficient for each genome embedding and set of closely and distantly related genomes (Fig. 2b, c). Only PST-TL methods, regardless of

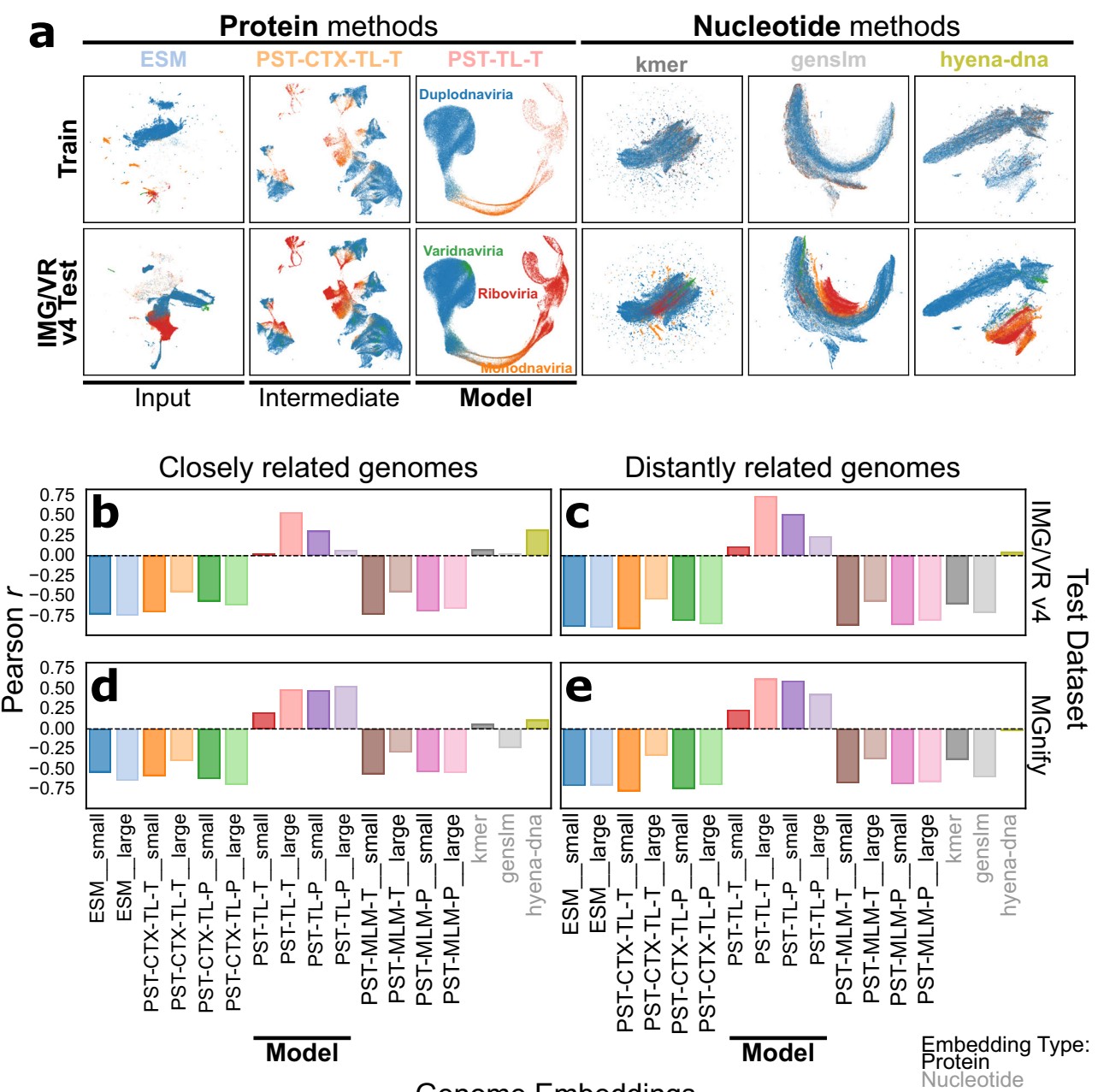

**Fig. 2 | PST learns biologically meaningful genome representations for diverse sets of viruses. a** UMAP dimensionality reduction plots for the genome embeddings produced by each method, color coded by the viral realm. "Kmer" represents 4-mer nucleotide frequency vectors. "PST-CTX-TL" methods are simple averages of the PST-TL protein embeddings over each genome. **b–e** Pearson correlation coefficients between the similarity in embedding space of the corresponding embedding and genome-genome similarity for the IMG/VR v4 (**b, c**) and MGnify (**d, e**) test datasets. Embedding similarity was calculated as angular similarity of L2-normalized embeddings. Genome-genome similarity is the harmonic mean of Average Structural Identity (ASI; see Methods) and the proportion of shared genes between each pair of genomes based on structural information. ASI was only computed between genomes that clustered together based on the corresponding embedding. Genome pairs were considered "closely related" (**b, d**) if there were any protein sequence alignments between any of the proteins from each genome. Inversely, genome pairs were considered "distantly related" (**c, e**) if there were no protein sequence alignments and only protein structural alignments between any of the proteins from each genome. The actual scatterplots can be found in Supplementary Figs. 9–12.

CV strategy or test dataset, displayed a positive correlation for both closely and distantly related genomes (Fig. 2b–e, Supplementary Figs. 9–12). Notably, PST-TL were also the only embeddings to show an increased correlation for the distantly related genomes compared to the similar genomes. This highlights the greater sensitivity in detecting genome-genome relationships for PST-TL models, especially for remote relationships. Inversely, kmer and HyenaDNA showed a positive correlation only for closely related genomes (Fig. 2b, d, Supplementary Figs. 9,11), which likely reflects the limited search range of

nucleotide-based methods. This is further supported by the sharp decrease in correlation for these two methods when evaluating the distant genomes (Fig. 2d, e, Supplementary Figs. 10,12). Interestingly, all other protein-based genome embeddings showed a negative correlation between the embedding similarity and ASI score, suggesting that these genome embeddings do not accurately capture genome-genome relationships (Fig. 2b–e, Supplementary Fig. 9–12). PST-CTX-TL genome embeddings, which are simple averages of PST protein embeddings compared to the learned weighted averages of PST-TL

genome embeddings, showed slightly less negative correlations but still performed poorly. Additionally, there was potentially representation collapse in the genome embeddings from PST-MLM models, most notably for PST-MLM-T, since there was little variation in embedding similarity (Supplementary Figs. 9–12). This arises even though PST-MLM models are not directly trained to produce genome embeddings (see Methods), unlike PST-TL models. Overall, these analyses suggest that full encoder-decoder PSTs are superior for genome representation learning.

We also evaluated the taxonomic purity of both the viruses and their hosts across the genome clusters, which did not strongly separate any method for either test dataset (Supplementary Figs. 13, 14). This may suggest that current viral taxonomy is not as informative for understanding viral-viral relationships across diverse sets of viruses compared to AAI and ASI, which are based on information more intrinsic to viral genomes. Further, this could also be skewed by the low proportion of viruses with a known or predicted host (Supplementary Fig. 2c).

## PST detects important viral protein functions

PST genome representations are produced as a function of the input protein embeddings that get contextualized by the intermediate PST encoder. Thus, we expected that biologically meaningful genome embeddings should be generated from relevant protein representations. We first analyzed the attention scores of each protein per genome from the large PST-TL models, which are used as importance scores when pooling the PST protein embeddings for the final genome representation. We considered that the general function of each protein was likely associated with high attention. Indeed, structural proteins (head, packaging, tail) and replication or nucleotide metabolism proteins were generally most attended to by the model, regardless of PST model or annotation database (Fig. 3a, b, Supplementary Fig. 15). This is intuitive since these proteins are essential to viruses and likely reflects their relatively greater abundance in the dataset (Supplementary Fig. 16). Further, we found a subtle association between the attention scores with the number of proteins belonging to the same sequence identity-based cluster (Supplementary Fig. 17). This reflects the model assigning a higher weight to proteins seen more frequently.

To assess the ability of PST to detect protein relationships, we conducted a similar analysis as with the genome clusters. The embedding-based protein clusters were generated using the Leiden algorithm on a similarity-weighted kNN graph with 15 neighbors and a similarity threshold of 0.9. This approach led to 1.5–2 M protein clusters for the IMG/VR v4 test set and 20–35 k protein clusters for the MGnify test set with average cluster sizes ranging from 2 to 6 proteins (Supplementary Fig. 18). For each dataset, ≥75% of proteins were clustered (Supplementary Fig. 18). We then performed a similar purity analysis of the protein clusters with respect to VOG and PHROG functional categories that did not strongly indicate which protein embeddings produced the most functionally pure protein clusters (Supplementary Fig. 19), although PST-TL-T_small, PST-TL-P_small, PST-TL-P_large, and GenSLM were consistently the best. The PST-MLM models tended to perform on par with ESM2 for this task.

## PST clusters related protein functions into function modules

To identify cases where PST outperforms the input ESM2 for protein tasks, we considered that genome context could impart a greater sensitivity for detecting evolution-driven genome organization. More specifically, we suspected that PST would be capable of identifying groups of associated protein functions that reflect this underlying genome organization. For example, the late genes encoding for structural, packaging, and lysis proteins are adjacent and transcribed by a single promoter in the Lambda genome[35]. We, therefore, assessed protein function co-clustering patterns. For each protein cluster, we calculated the number of times pairs of proteins belonging to different PHROG functional categories co-clustered against the number of times each pair of categories would be expected to co-cluster based on the underlying distribution of the PHROG database categories. The resulting enrichment networks showed that PST-TL could group proteins based on broader function modules in the IMG/VR v4 test set (Fig. 3c, Supplementary Fig. 20). For example, tail, head and packaging, connector, and lysis proteins, which are notably late gene proteins, consistently co-clustered above background in PST-TL protein clusters. Additionally, DNA-interacting (nucleotide metabolism, lysogeny, and gene expression) function modules were enriched in PST-TL-P and GenSLM protein clusters. Interestingly, no interpretable modules were detected when using ESM2 protein embeddings (Fig. 3c, Supplementary Fig. 20). Moreover, only PST-MLM-T_large detected the late gene and DNA-interacting modules. Since all PST-TL models but only one PST-MLM model were able to detect these modules, this suggests that the capacity to identify functional modules is specific to PST-TL and not PST-MLM. Notably, these functional modules were not detected by any method in the MGnify test set (Supplementary Fig. 21). This could be due to the ~20x reduction in the total number of annotated proteins in the MGnify set relative to the IMG/VR v4 set that prevent accurately identifying these co-clustering relationships. However, this could also reflect evolutionary differences of soil viruses compared to viruses from other ecosystems.

These results were also consistent with the proportion of protein clusters from the IMG/VR v4 test set that we considered as belonging to these function modules, as PST-TL showed the highest proportions (Fig. 3d–g, Supplementary Fig. 22a, c). For late genes and when using VOG annotations, there were not significant differences between the protein embeddings with the MGnify test set (Fig. 3e, Supplementary Fig. 22b, d). However, PST-TL showed a higher proportion of DNA-interacting clusters (Fig. 3g), despite this module not being detected in the co-clustering networks (Supplementary Fig. 21). This difference is likely because this approach only considered "pure" clusters that represent these functional modules, while the previous strategy quantified all clusters without considering an overall cluster summary. Nonetheless, these data demonstrate that considering genome context better enables PST to detect broader functional associations implicitly encoded in viral genome organization.

## PST expands annotation for proteins of unknown function

Interestingly, hypothetical proteins unable to be annotated by either the VOG or PHROG databases were generally considered the most important by PST-TL models (Fig. 3a, b, Supplementary Fig. 15). One explanation is that since proteins of unknown function make up 70-90% of all proteins in the PST test datasets (Supplementary Fig. 16, Supplementary Data 2), it is likely that there are true viral structural and replication proteins that have diverged at the sequence level among the unannotated proteins. Since PST-TL embeddings showed positive correlation with a structure-based genome-genome similarity metric (Fig. 2b, c, Supplementary Figs. 9–12), we investigated if PST could identify structural similarity more directly. To test this, we selected unannotated proteins that cluster with detectable capsid proteins to check if these unannotated proteins contained conserved capsid-like structural folds that were missed by sequence similarity searches. We filtered the proteins from the test viruses to maintain proteins belonging to protein clusters that contained only annotated capsid proteins or hypothetical proteins. We then used foldseek[36] and ProstT5[37] to translate this protein set into a structural alphabet for searching against Protein Data Bank[38] structures for structural homology. To validate the structural reasoning of this approach that does not directly infer a protein structure, we independently aligned the structures of the reference HK97 major capsid protein[39] with two different AlphaFold 3-predicted[40] structures using the most structurally similar proteins from our dataset: one that was detected by a VOG profile with unknown function (Fig. 4a) and one undetected entirely

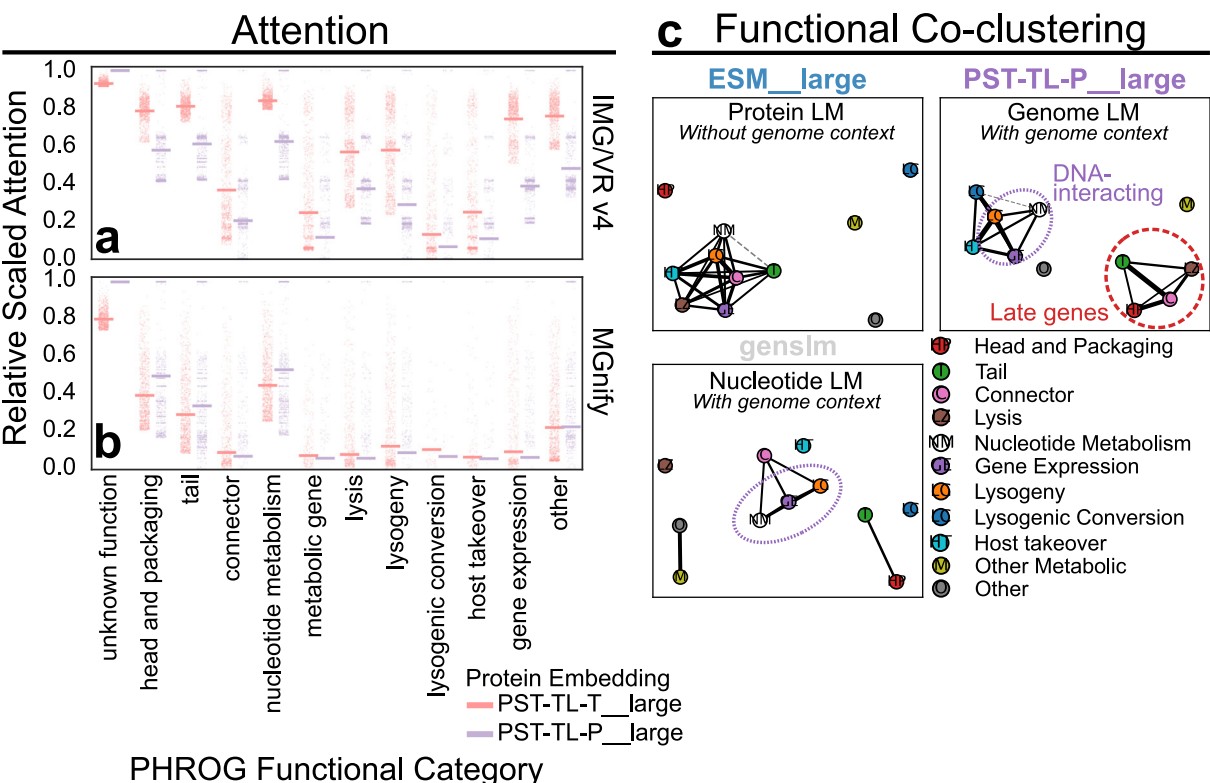

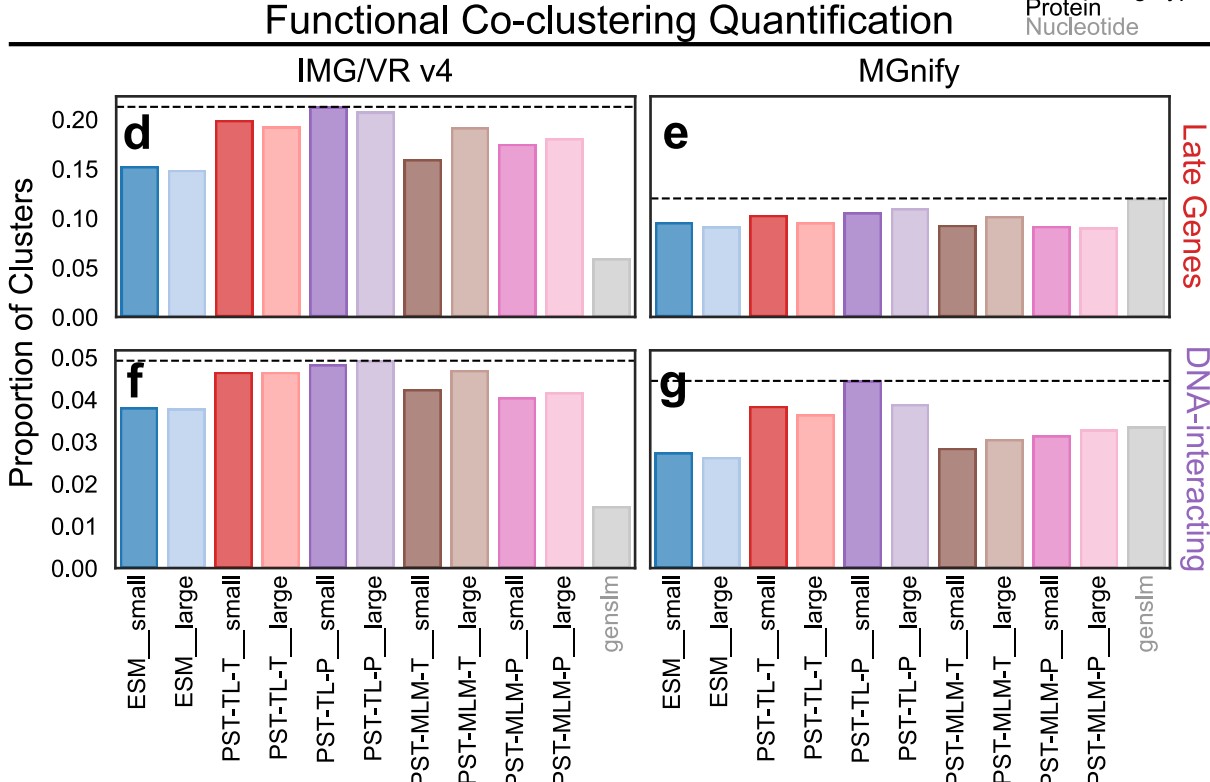

(Fig. 4b). The strong alignments indicate that our workflow can accurately identify capsid-fold-containing proteins from the protein sequence alone. Using this approach, the PST-TL models generally showed the greatest proportion of unannotated proteins with structural homology to known capsid proteins, regardless of CV strategy (Fig. 4c, d). GenSLM ORF embeddings were also better than other protein embeddings for this task, likely due to being pretrained on microbial genomes, which would contain some viral sequences, and finetuned on SARS-CoV-2 genomes. The proportion of unannotated proteins clustering with capsid proteins was consistently the lowest with ESM2, PST-MLM, and, surprisingly, PST-TL-T_large protein embeddings.

We next considered that similarity in embedding space could be used to propagate functional labels from annotated to unannotated

**Fig. 3 | PST leverages genomic context to learn protein function relationships.** **a, b** Relative scaled attention from large PST-TL models for proteins belonging to different PHROG functional categories for IMG/VR v4 (**a**) and MGnify (**b**) test datasets. The per-protein attention scores from each model were normalized by the number of proteins encoded per scaffold (see Methods). For each model, these normalized attention values were then rescaled so that the maximum value was 1. Then the top 1000 proteins based on the rescaled normalized attention were chosen for each functional category (i.e., $n = 1000$). The dots are the values for one of the 1000 top attended proteins from each category, and the horizontal bar is the mean. **c** Summary of functional co-clustering based on PHROG annotations of the IMG/VR v4 test set. Each connected component was clustered in a co-occurrence graph using the Leiden algorithm with resolution of 1.25. The edges indicate pairs of functional categories that were more enriched in protein clusters defined by clustering the k-nearest neighbor graph of the corresponding protein/ORF proteins. embeddings relative to the background distribution of annotation profiles. The length of the edges reflects the degree of enrichment since the networks were visualized using a spring force algorithm. Dashed gray lines indicate connections that were less enriched than or equal to expected, while solid lines were more enriched than expected. **d–g** The proportion of protein clusters that correspond to Late Gene (**d, e**) or DNA-interacting (**f, g**) modules for the IMG/VR v4 (**d, f**) and MGnify (**e, g**) test datasets. A protein cluster was considered to represent a functional module if all proteins annotated by PHROG in the cluster belonged to the subcategories that composed each module (Late Genes: Head and Packaging, Tail, Connector, Lysis; DNA-interacting: Nucleotide Metabolism, Lysogeny, Gene Expression). Each protein cluster was also required to have annotated proteins belonging to at least 2 of the subcategories. The dashed lines mark the method that has the greatest proportion of protein clusters corresponding to the indicated functional module.

proteins. To evaluate the annotation transfer ability of PST, we checked if any of the nearest neighbors in embedding space were annotated (Fig. 4e, g). We further constrained this search to require all neighbors belong to the same VOG functional category as an additional source of confidence for annotation transfer (Fig. 4f, h). These results showed little difference for most methods, except for GenSLM, PST-TL-P_small, and PST-TL-T_small consistently being the worst. It is possible that the smaller embedding sizes for PST-TL-P_small and PST-TL-T_small may have decreased the performance since the larger models perform better. As additional neighbors were searched, the improvement proportion for all protein-based methods tended to decrease after >6 neighbors were evaluated and required to have the same functional category (Supplementary Fig. 23b, d). Meanwhile, GenSLM showed plateauing improvement when requiring congruent functions among the nearest neighbors (Supplementary Fig. 23b, d). This is consistent with nucleotide sequences having a lower evolutionary range compared to protein sequences and structures, as GenSLM ORF embeddings could not generally identify any additional annotated proteins (Supplementary Fig. 23). Further, we already demonstrated that broader functional relationships can be detected by PST-TL protein embeddings, which explains why protein-based representations would decrease in performance as more neighbors are considered. This, however, indicates an initial grasp of both specific and broad protein function that could be tuned for different downstream tasks. Overall, these results indicate that protein embeddings are better for annotation transfer.

**PST can be applied toward viral-host prediction**

Since we expect that PST can be used as a general-purpose model for downstream viromics tasks, we used the large PST genome embeddings for viral-host prediction as a proof-of-concept. We adapted a graph framework described previously[41] that models this scenario as a link prediction task in a virus-host interaction network. Briefly, the objective is to predict for any pair of virus and host whether there should be a link, indicating a prediction for infection of that host by the corresponding virus (Fig. 5a).

We implemented a variant of the GNN-based CHERRY algorithm[41] (Fig. 5a), swapping out the node (genome) embeddings of both viruses and hosts with either ESM2, various PSTs, or the tetranucleotide frequency (kmer) vectors that CHERRY uses. Although this design is likely suboptimal for PST models, which have embeddings specialized for viruses but not hosts, it enables a direct comparison of the choice of genome embedding instead of various virus-host genome embedding combinations. We then trained these models using the training dataset of the host prediction tool iPHoP[42] to compare with previously published work (Supplementary Fig. 24). Then, each trained model and iPHoP were evaluated using the same iPHoP test dataset. We assessed whether the true host species for each test virus could be identified with high confidence (Fig. 5b). The model using PST-TL-T genome embeddings significantly outperformed all other methods at the host species level, even previously published tools (Fig. 5b). Notably, PST-TL-P did not perform as well as PST-TL-T, despite generally performing as well as PST-TL-T in previous tasks. Conversely, PST-MLM-P outperformed PST-TL-P moderately at the highest confidence threshold and substantially at the lower threshold (Fig. 5b), which is surprising since PST-MLM-P has not performed well in the prior evaluations. This could be explained by PST-MLM-P and PST-TL-T having 10x more trainable parameters compared to PST-TL-P (Supplementary Table 4). PST-MLM-T, in contrast, did not make any correct predictions that met the confidence requirements and has 50% fewer parameters. Another potential source of bias that could skew the results generally in favor of PST embeddings is that there are viruses in the iPHoP test set that are similar to those in the PST training set (Supplementary Fig. 25a). However, excluding these viruses did not change the overall results (Supplementary Fig. 26).

## Discussion

Here, we present the PST framework for modeling genomes as sets of proteins, where each protein is initially represented by information-rich ESM2 protein embeddings. PST contextualizes the input protein embeddings and subsequently yields genome representations as weighted averages of contextualized protein embeddings, which can be targeted toward either protein-level or genome-level downstream tasks. When pretrained on a large, diverse dataset of viral genomes, PST-TL demonstrated superior ability in detecting evolutionary relationships among viral genomes (Fig. 2b–e). Notably, encoder-only PSTs like PST-MLM and PST-CTX-TL did not show any correlation between genome embedding similarity and ASI (Fig. 2b–e). This suggests that a trainable decoder that learns a weighted average of protein embeddings is required for optimal genome representation. Further, our PST-MLM models were trained in a similar way to a previous genome language model[10]. The poor performance of PST-MLM models in both genome and protein tasks indicates that MLM is possibly not the best training regime for genome language modeling, especially for highly diverse genomes like viral genomes.

At the protein level, the PST-TL protein embeddings demonstrated patterns of broad function grouping, consistently clustering late gene proteins together (Fig. 3c–g). Additionally, PST-TL often clustered capsid-fold-containing proteins that could not be annotated by VOG with annotated capsid proteins (Fig. 4c, d). This suggested, alongside a positive correlation between embedding similarity and protein structure-based genome-genome similarity (Fig. 2b–e, Supplementary Figs. 9–12), that PST-TL uses inferred structural information for relating proteins and, subsequently, genomes. PST-TL further showed high sensitivity for annotation transfer (Fig. 4e–h). Performance for these protein-level tasks could be further improved by finetuning the ESM2 pLM with viral sequences and by training a PST with a dual objective that more directly considers protein-protein and genome-genome relationships. The performance of PST-TL-P_large, specifically, across these tasks is notable since this model has ~10% the

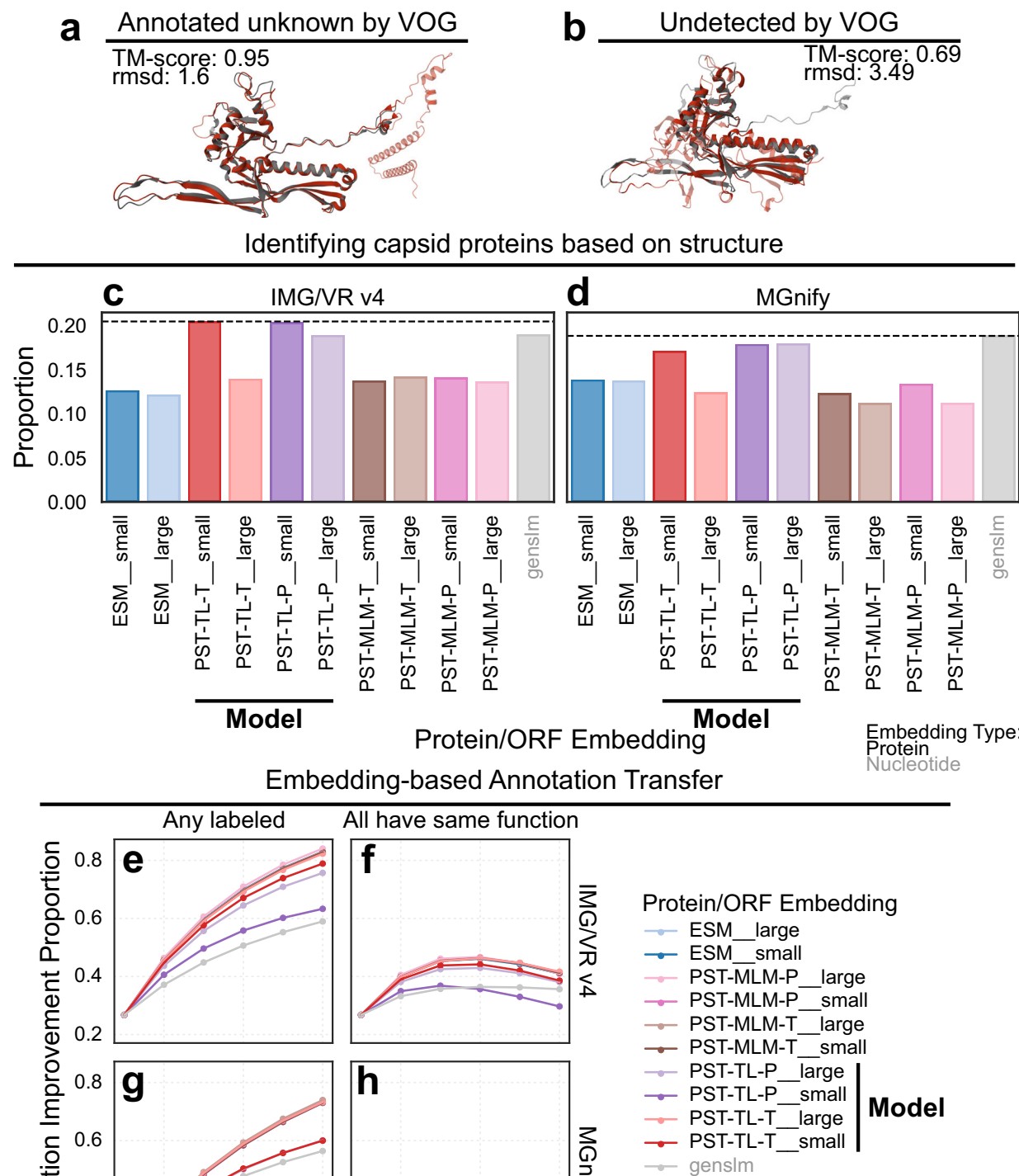

number of parameters compared to the other large PST models (Supplementary Table 4). While data leakage from the ESM2 UniRef50 pretraining data could inflate the performance of PST, PST consistently outperformed ESM2 on the protein tasks evaluated, suggesting the ESM2 pretraining data was not the sole source of performance.

Finally, when applied toward a viral-host prediction task, the PST-TL-T genome embeddings were able to detect the true host species for the greatest number of viruses when compared against two previously published host prediction tools (Fig. 5b). We notably refrained from overanalyzing the subtle differences in performance in the proof-of-concept host prediction task since there are numerous training techniques beyond the scope of our work that could have resulted in a superior PST-based host prediction model. For example, the PST genome embeddings are likely not well-suited for host genome

**Fig. 4 | PST expands functional annotation of hypothetical proteins. a, b** Structural alignments with the HK97 major capsid protein (PDB: 2FS3, gray) for a protein annotated by VOG as unknown (**a**, "IMGVR_UViG_3300038749_000016 | 3300038749 | Ga0423190_00012 | 260998-304157_27") and another undetected by VOG (**b**, "IMGVR_UViG_2687453601_000002 | 2687453601 | 2687454426 | 1161639-1205965_50"). The red cartoon diagrams are the query proteins from our dataset and were chosen due to being the most similar to the HK97 capsid protein (2FS3) from each category. **c, d** The proportion of proteins from the IMG/VR v4 (**c**) and MGnify (**d**) test datasets unannotated by VOG clustering with annotated capsid proteins that have detectable structural homology with known capsid folds. Structural homology was detected using foldseek searching against the Protein Data Bank database. **e–h** The proportion of proteins unannotated by VOG whose nearest neighbors in embedding space are annotated. The colors indicate the protein/ORF embedding. Nearest neighbors were searched using angular similarity after L2-normalizing the protein embeddings. **e, g** A hit was considered for each unannotated protein if any of the neighbors less than or equal to the current number of nearest neighbors were annotated. **f, h** A hit was considered similarly to (**e, g**) with the additional constraint that all of the current set of nearest neighbors must belong to the same VOG functional category. Unannotated proteins were not used to penalize the score. The rows indicate the test set used: IMG/VR v4 (**e, f**) and MGnify (**g, h**).

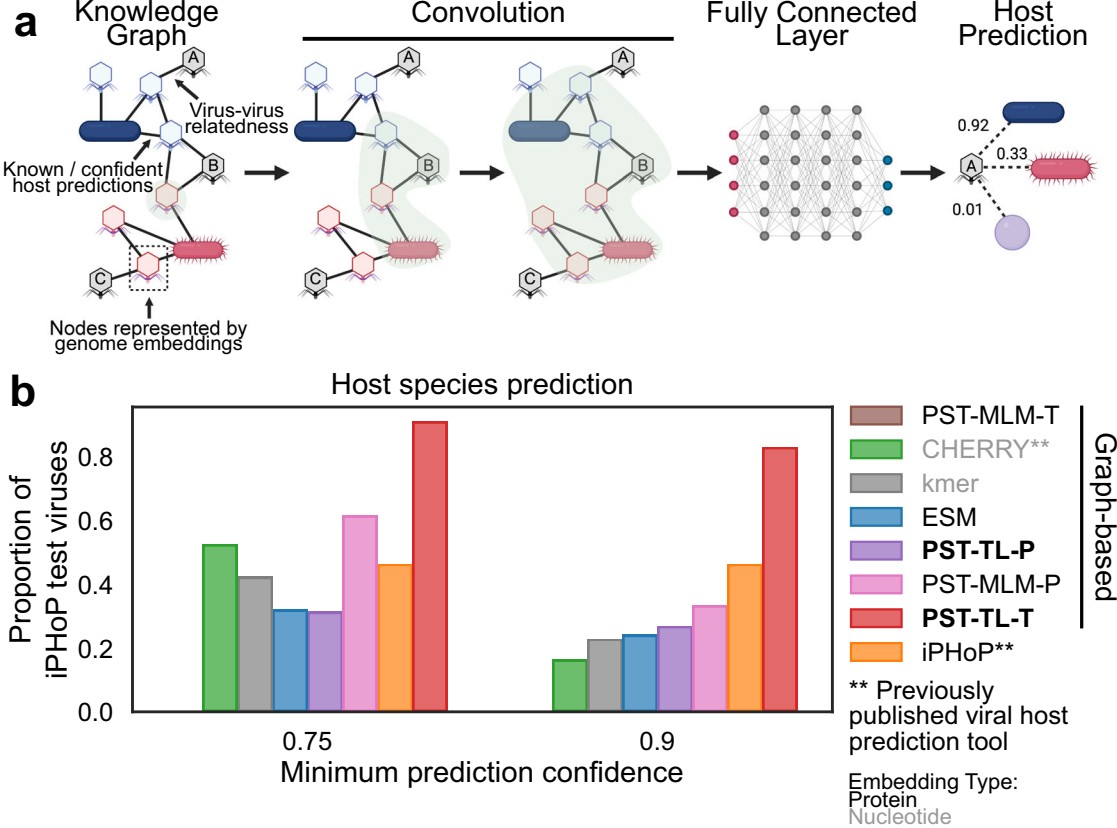

**Fig. 5 | PST improves host prediction. a** Graph neural network approach for host prediction developed by CHERRY. The node representations are swapped out to the corresponding data type. **b** The proportion of iPHoP test viruses whose true host species is predicted above the indicated confidence threshold. None of the 1636 iPHoP test viruses were filtered for similarity to those in the PST training set. The graph-based models were trained in this study, while "iPHoP" represents the output of running the iPHoP tool on the same test set. All PST and ESM models use the "large" versions for genome embeddings. PST-MLM-T did not have any predictions above the indicated confidence thresholds.

representation. It is, therefore, important to emphasize that the PST-based host prediction model performed on par with (and sometimes better than) existing host prediction tools without PST being initially tasked with host prediction and without significant training time.

It is imperative to reiterate that this superior performance in a variety of viromics tasks emerged despite not training PST with these objectives. Taken together, our results indicate that PST is suitable as a foundation model for common viromics tasks, such as virus identification, taxonomy, host prediction, protein annotation, genome binning, etc. (Fig. 6a). We anticipate that more thorough studies for downstream viromics problems will benefit from starting from our pretrained PST models. Additionally, finetuning PST can bring even greater performance for these downstream tasks. For example, fine-tuning an end-to-end host prediction model with a virus-host dataset would likely significantly improve predictive power compared to what we observed. Further, the iPHoP training dataset has limited diversity

(Supplementary Fig. 25b), which could suggest that the results here are not representative of true performance on less represented viruses. Nonetheless, our work has provided a preliminary framework for a standalone PST-based host prediction tool.

There has been increasing discussion around the biosecurity risks posed by biological foundation models due to potential biosecurity threats, such as generating novel pathogenic viruses or guiding gain-of-function viral mutations[43,44]. For example, the AlphaFold 3 web server does not allow predictions for certain viral proteins[40], Evo excluded viruses with eukaryotic hosts from its pretraining data[45], and ESM3-open filtered viral sequences and select agents from its training sets[46]. While developing PST, we have assessed the ethical implications of this viral foundation model and perceived it to have a low biosecurity risk, in particular with respect to human pandemics. First, only 0.2% of the PST training viruses infect humans. Of these, only four are on the CDC's list of bioterrorism agents (https://emergency.

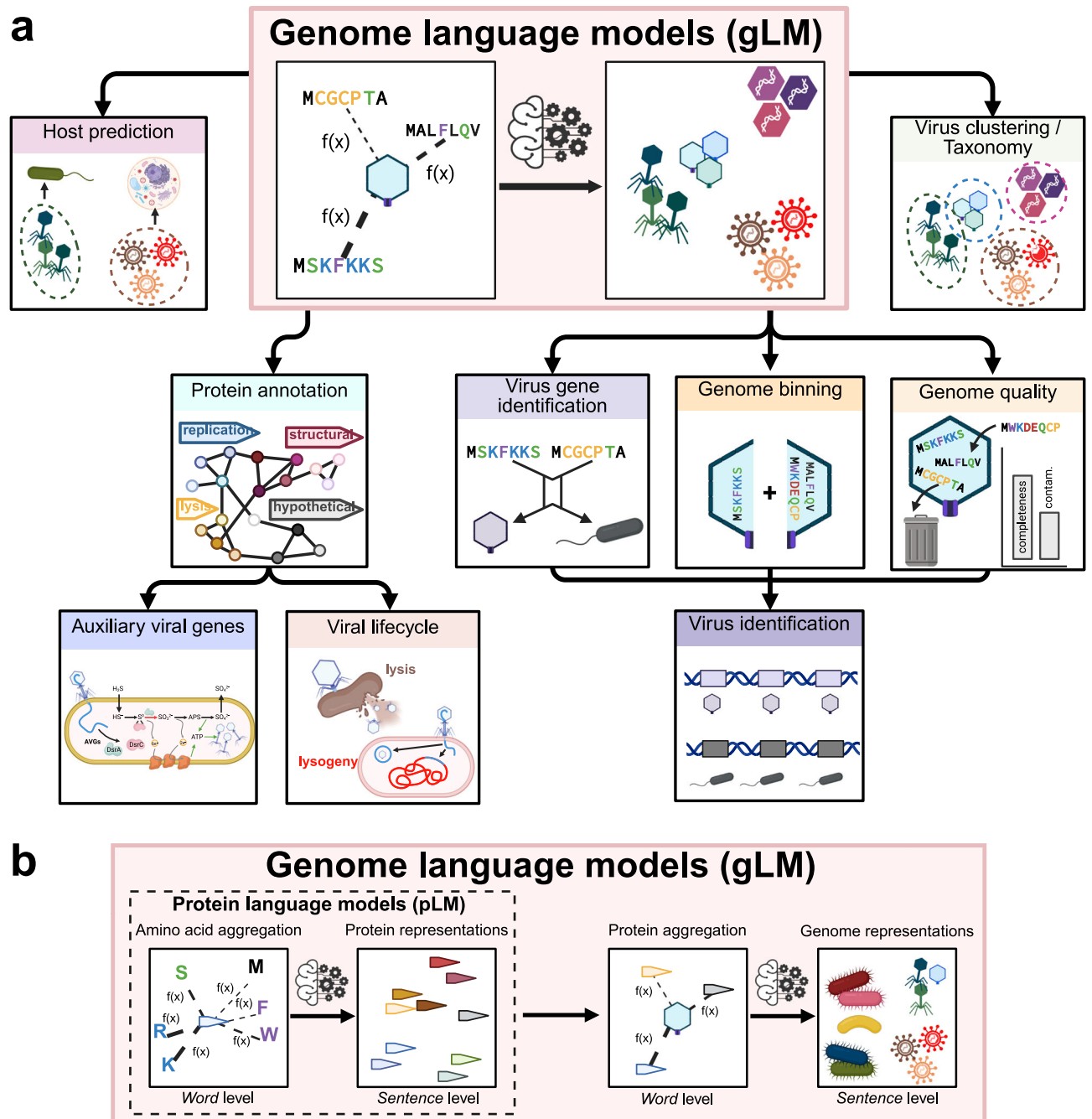

**Fig. 6 | PST can be a general-purpose microbial and viral genome language model. a** Potential downstream tasks of the pretrained PST that represent commonly desired steps of a typical computational viromics pipeline. **b** Example workflow of a genome language model based on PST that could incorporate both microbial and viral input genome datasets.

cdc.gov/agent/agentlist-category.asp; Filoviridae viruses: Ebolavirus and Marburgvirus), and ten more are under surveillance by the National Respiratory and Enteric Virus Surveillance System (https://www.cdc.gov/nrevss/php/dashboard). Further, only 1% of the training viruses infect mammals, which would be the most likely viral reservoirs that could spillover into human populations. Since our model was not trained considering host identity, the low abundance of these viruses in the training dataset likely minimizes their influence on the learned PST embeddings. Second, the lowest resolution of PST is at the protein level, meaning that it would be difficult to reverse engineer a de novo viral genome using PST. While a nucleotide language model reported the ability to generate de novo bacterial virus genomes[33], the similarity of these genomes to the training dataset was not investigated, so it is likely

that they do not differ substantially from the training data. Reverse engineering genomes from our protein-based work is further complicated by the complexities of how human viruses tend to encode and express genes (overlaps, alternative starts, alternative splicing, post-translational processing, etc.). These molecular biology issues likely mean that achieving in vivo activity of a generated viral genome would be challenging as previously discussed[47].

We additionally consulted with three experts in biological machine learning and data science to assess the potential biosecurity risk of our work before releasing the PST code and model weights. There was unanimous agreement that PST offered potential benefits for virology and other biological fields, leading to full support for publicly releasing the source code and trained models. Two reviewers

did not perceive a relatively greater threat by PST compared to the ESM2 model used as input. The other had no general concerns but acknowledged that there could be unforeseen issues. Therefore, the external evaluation is consistent with our internal assessment that the demonstrated and potential future benefits (Fig. 6a) of our work to advance our understanding of viruses outweigh any hypothetical threats that would require significant resources to unleash.

Finally, although trained on viral proteins and genomes, our PST architecture is agnostic to the source of the proteins and type of genomes. The only requirements of our framework are the ordered protein sequences and genome strand of each ORF. These requirements are more easily satisfied by microbial genomes, where computational ORF calling is both accurate and common. However, PST could theoretically work with large enough datasets of experimentally determined ORFs from eukaryotes. Nonetheless, we propose that our PST implementation is equally appropriate for developing a microbial foundation model to solve similar challenges in microbial genomics (Fig. 6b), which notably also include poor protein annotation rates and high sequence divergence. In fact, our foundation PST-TL-T model was still useful for host genome representations in the virus-host prediction task (Fig. 5), despite only being trained on viruses.

## Methods

### Viral genome datasets
We acquired viral genomes from 12 different publicly available sources[1,19–29] as a training dataset. For GTDB (r202), we used PhageBoost[48] (v0.1.7) with default settings to identify integrated proviruses, filtering predictions that did not encode at least 20 proteins. We then filtered genomes that were not considered complete or high-quality as defined by CheckV[49] (v1.0.1). We then dereplicated this set of genomes using a custom workflow. We first used skani[50] (v0.1.0 sketch: --fast) to compute pairwise average nucleotide identity (ANI) between all pairs of viruses. We constructed a graph where edges connected viruses with ≥95% ANI and ≥50% genome coverage of the alignment for both genomes. The edge weights were the product of ANI and coverage. We then used the Markov clustering algorithm[51] (mcl v14-137 -I 2.0) to cluster this graph, taking one genome from each cluster at random as a representative genome. This resulted in 103,589 viral genomes for training.

We then chose the most complete, least contaminated, and longest genome for each viral operational taxonomic unit in IMG/VR v4[12], ensuring that each representative was considered high-quality by CheckV, as one test dataset. We then dereplicated this putative test dataset with the training dataset using a similar approach as above with skani (--slow, ≥95% ANI, ≥85% coverage) and mcl. We kept all viruses that did not cluster with training viruses. For both datasets, we filtered out viruses predicted to encode only 1 protein. This resulted in 151,255 genomes spanning 182,906 scaffolds.

We curated a second test dataset composed solely of soil viruses, which are significantly less represented in the IMG/VR v4 test dataset (15.3% compared to 49.7% aquatic viruses), identified from soil metagenomes deposited in MGnify[13]. We downloaded 145 soil metagenomes with assemblies available up to January 10, 2025, filtered for contigs ≥2.5 kb in length, and used geNomad[52] (v1.7.4) to identify viral scaffolds from the assemblies. This resulted in an initial set of 17,802 viruses. Then, we dereplicated these viruses using the same skani-mcl workflow with both the training and IMG/VR v4 test datasets. This led to a final total of 12,857 MGnify test viruses.

For all viruses, we predicted protein ORFs using the Python bindings of prodigal called pyrodigal[53] (v2.3.0) for single-contig viruses and prodigal-gv (v2.11.0) for viral metagenome-assembled genomes (vMAGs). We did not consider the updates made by prodigal-gv[52] (added gene models for giant viruses and viruses using alternative genetic codes) to be substantial enough to apply to the entire dataset given the scale and distribution of the data. With these protein predictions, we filtered out viruses predicted to encode only one protein

(the final number of genomes listed above reflects this final filtering step). The total number of proteins encoded in each dataset were 6,391,562 for the training dataset, 7,182,220 for the IMG/VR v4 test dataset, and 141,643 for the MGnify test dataset.

For the training viruses and MGnify test viruses, viral taxonomy not provided by IMG/VR v3 was assigned using geNomad[52] (v1.7.2) to get labels that were consistent with the current standards. For the IMG/VR v4 test viruses, we used the provided taxonomic labels since they were consistent with current standards, and most were predicted using geNomad also. We did not perform host prediction on the training or IMG/VR v4 viruses, since that was already provided by the viral-centric source databases. For the MGnify test viruses, however, we used iPHoP (v1.3.3) to predict hosts, since we detected these viruses in this study. The summary of information for the training and test viruses can be found in Supplementary Data 1.

### ESM2 protein language model embeddings
PyTorch (v2.1.0)[54] and fair-esm[3] (v2.0.0) were used to obtain protein embeddings. We refer to the ESM2 models "esm2_t6_8M_UR50D" (six layers, 8 M parameters, 320-dimensional embedding) and "esm2_t30_150M_UR50D" (30 layers, 150M parameters, 640-dimensional embedding) as "ESM_small" and "ESM_large", respectively. The amino acid embeddings in each protein were averaged for a single $d$-dimensional vector. For proteins longer than 20,000 amino acids, the sequence was split in half, and the embeddings for each half were then averaged for the final embedding. This only affected one bacterial protein in the host prediction analysis.

Generating the fixed ESM2 embeddings is the most computationally expensive step for the entire PST model, but this only needs to be done once. The approximate inference rate across each dataset used here was 614.28 proteins/second for ESM_small and 130.80 proteins/second for ESM_large using one A100 GPU. This led to the following inference times for ESM_small (training set: 2.89 h, IMG/VR v4 test set: 3.25 h, and MGnify test set: 3.84 min) and for ESM_large (training set: 13.57 h, IMG/VR v4 test set: 15.25 h, and MGnify test set: 18.05 min). Notably, this step can scaled up to the number of available GPUs since the ESM2 embeddings are independent for each protein, meaning that input files can be split evenly and assigned to each GPU for parallelized inference.

### PST model architecture
PST was built using PyTorch (v2.0.0), PyTorch Geometric[55] (v2.3.1), and PyTorch-Lightning (v2.0.7). PST draws inspiration from deep learning of set-structured data like the Set Transformer[14] while using modifications that are specific to pointsets[15], which are sets whose items are $d$-dimensional vectors. PST, thus, models genomes as a set of proteins $G_i$ where $g_{ij} \in G_i$ is the $j$th protein in the $i$th genome. Each protein is initially represented by its $d$-dimensional ESM2 protein embedding, with $G_i \in R^{n_i \times d}$ where $n_i$ is the number of proteins encoded in genome $G_i$. We did not finetune the ESM2 models, so the ESM2 embeddings were used as frozen inputs. For each protein $g_{ij}$, learnable embeddings for both the position in the genome and for the encoding strand were concatenated to the ESM2 embeddings. The positional embeddings for proteins were relative to the positions of the proteins in each genome and are used so the model learns relative ordering of proteins. For fragmented genomes, such as vMAGs, the protein ordering was only done relative to each scaffold instead of the entire set of scaffolds that compose a fragmented genome.

To account for the large variation in the number of proteins encoded by each genome, we used a memory-efficient graph-based implementation that considers each genome as a graph and each protein as nodes in the genome graph. Notably, each individual genome matrix $G_i$ is stacked for each minibatch, so there is no padding. Then, an indexing pointer keeps track of the offsets (number of rows/ proteins) for each genome for efficient access. For memory efficiency

and to model real fragmented genomic data, we break each genome graph into subgraphs whose node sets include 15–50 mutually exclusive, contiguously located proteins. The size of each subgraph was tuned and is, thus, fixed. These nodes are all fully connected in each subgraph such that all proteins in each genome subgraph attend to each other in the PST encoder. We prevent subgraphs with 1 node by adding possible singleton node cases to the previous subgraph. The subgraph size ("chunk size") hyperparameter is constant for all genomes. Thus, a minibatch of $N$ genomes is represented by a single graph $\mathcal{G} = (X, E) = \text{Stack}(G_i)$ where $X$ is the total number of proteins encoded by the $N$ genomes. $E$ is the total number of protein-protein edges and is a function of the subgraph size and number of proteins per genome.

PST uses an encoder-decoder paradigm previously described with the Set Transformer[14]. The encoder uses multi-head self-attention to contextualize each protein by the other proteins within the same scaffold (i.e., a single contiguous nucleotide sequence). Then, the decoder uses multi-head attention pooling to summarize the genome as a weighted average of contextualized protein embeddings. To contextualize the proteins in each genome, we used a graph-based implementation of multi-head scaled-dot product self-attention[16] in each layer of the PST encoder:

$$\alpha_{ij} = \text{GraphSoftmax}\left(\frac{\left(W^Q x_i\right)\left(W^K x_j\right)}{\sqrt{d}}\right) \qquad (1)$$

$$\text{MultiHeadAttn}(\mathcal{G}) : x_i^{(l+1)} = W^{Q,(l)} x_i^{(l)} + \sum_{j \in \mathcal{N}(i) \cup \{i\}} \alpha_{ij}^{(l)} W^{V,(l)} x_j^{(l)} \qquad (2)$$

where $x_i^{(l)}$ is the embedding vector for the $i$th protein at the $l$th encoder layer, $d$ is the protein embedding dimension. Likewise, $W^{\cdot,(l)}$ is the weight matrix for the query, key, and value at the $l$th encoder layer. $\mathcal{N}(i)$ is the set of protein neighbors for the $i$th protein in the same genome subgraph. $\alpha_{ij}$ is the scaled-dot product attention calculation. GraphSoftmax is a modified softmax function that only normalizes the attention values within the set of subgraphs that belong to the same genome. Thus, only proteins in the same subgraph attend to each other, but the attention values are normalized by all proteins in the genome. To enable multi-head attention, we split the input protein embeddings in the same number of chunks as the number of attention heads along the embedding dimension. After the self-attention calculation, we concatenate the outputs from each head back together. Further, we followed a pre-normalization strategy in which we normalized the input protein embeddings before the linear layers. Specifically, we used the GraphNorm (implemented in PyTorch-Geometric) normalization operator that normalizes the protein embeddings only within each genome. Additionally, we used the corresponding skip connections, in which untransformed inputs are added to values post-attention. A full PST encoder layer can thus be mathematically represented as the following set of equations:

$$
\begin{aligned}
X^1 &= \text{GraphNorm}\left(X^0\right) \\
X^2 &= \text{MultiHeadAttn}(\mathcal{G}) \\
X^3 &= X^0 + X^2 \\
X^4 &= \text{GraphNorm}\left(X^3\right) \\
X^5 &= \text{FF}\left(X^4\right) \\
X^6 &= X^3 + X^5
\end{aligned} \qquad (3)
$$

where $X$ represents the intermediate protein representations and $X^0$ is the input protein embeddings in the stacked batch. FF represents a 2-layer feedforward network with Gaussian error linear unit (GELU[56])

activation and dropout after each layer. After the full PST encoder, a final GraphNorm operation was applied.

The PST decoder uses multi-head attention to compute a per-protein attention score to be used as the weights for a weighted average of protein embeddings over each genome. As described previously[14], multi-head attention pooling uses a learnable $d$-dimensional seed vector $S$ as the query when computing attention. During the attention calculation, the contextualized protein embeddings $X^C$ output from the PST encoder are projected onto $S$:

$$\text{Attn}\left(X^C, S\right) = \text{GraphSoftmax}\left(\frac{\left(W^Q S\right)\left(W^K X^C\right)}{\sqrt{d}}\right) \times \left(W^V X^C\right) \qquad (4)$$

The attention values from this projection are used to weight $X^C$. After re-weighting, $X^C$ is averaged over each genome to produce the final genome outputs. The full set of PST decoder equations is similar to the encoder (Eq. 3):

$$
\begin{aligned}
X^0 &= \text{GELU}\left(W X^C\right) \\
X^1 &= \text{GraphNorm}\left(X^0\right) \\
X^2 &= \text{Attn}\left(X^1, S\right) \\
X^3 &= X^0 + X^2 \\
X^4 &= \text{GraphNorm}\left(X^3\right) \\
X^5 &= \text{FF}\left(X^4\right) \\
X^6 &= X^3 + X^5 \\
X^7 &= \text{GraphPool}\left(X^6\right) \\
X^G &= \text{FF}\left(X^7\right)
\end{aligned} \qquad (5)
$$

where $W$ is the weights of a linear layer. GraphPool is a pooling (mean) operator over each genome graph that averages the contextualized weighted protein embeddings for each genome. Each FF is a different 2-layer feedforward network with GELU activation and dropout after each layer. $X^G$ is the final genome embeddings. See Supplementary Fig. 1 for a pictorial representation of the PST architecture.

For fragmented genomes like vMAGs, there is an additional step. Here, $X^G$ instead represents a scaffold-level embedding over the proteins encoded on that scaffold. This arises since PST contextualization and decoding only happen over contiguous sequences. To then produce a genome-level embedding, we use mean pooling over scaffold embeddings for all scaffolds composing a fragmented genome.

Inference using 1 A100 GPU takes between 7e-4–0.027 s/genome or 0.018–0.159 s/genome using 128 CPUs (AMD EPYC 7H12). When using 1 GPU, the total inference time for each dataset ranges from 30s–1.13 h depending on the number of genomes and proteins encoded per genome. For 128 CPUs, the total inference time takes significantly longer but is still reasonably between 40 s for the smaller MGnify dataset (12,857 genomes encoding 141,643 proteins) with ESM_small embeddings up to 6.68 h for the larger IMG/VR v4 dataset (151,255 genomes encoding 7,182,220 proteins) with ESM_large embeddings. PST can load datasets entirely into memory, so memory usage during inference will include the size of the dataset plus processing. However, there is an option to lazily load data from disk, which would increase processing times slightly. For the MGnify test set, which is more reminiscent of a typical dataset, this would be 170–339 MB for loading the dataset and an additional 942 MB–2.98 GB for the model (total of 1.1–3.3 GB). For larger datasets like the training and IMG/VR v4 test sets, the primary spike in memory usage would be from loading the dataset into memory (7.8–17 GB), but would also increase by the size of the dataset. To account for varying sizes of data,

we have implemented lazy loading as the default option during inference so that the major memory requirements are from the model, meaning that maximum memory should be under 3GB for the largest models. This can be further reduced by lowering the batch size, which is less important for inference.

## Training the PST foundation models with TL

The foundation viral PST models were trained using a self-supervised TL objective $\mathcal{L}(\mathcal{G})$ as described previously[15]:

$$D(G^a, G^p, G^n) = \left[ \left\| f(G^a) - f(G^p) \right\|_2^2 - \omega_i \left\| f(G^a) - f(G^n) \right\|_2^2 + \alpha \right]_+ \quad (6)$$

$$\mathcal{L}(\mathcal{G}) = \frac{1}{2N} \sum_{i=1}^{N} C_i \left[ D(G_i^a, G_i^p, G_i^n) + D(G_i^a, G_i'^p, G_i'^n) \right] \quad (7)$$

where $G_i^a$ is the $i$th genome treated as an anchor point, $G_i^p$ is the positive genome for the $i$th genome, $G_i^n$ is the negative genome for the $i$th genome, and $G_i'$ is the augmented genome for the $i$th genome created using the PointSwap sampling method[15]. $f(\cdot)$ is the function modeled by the full PST neural network, and $||x - y||_2^2$ is the squared L2 (Euclidean) distance between the vectors $x$ and $y$. $C_i$ is an optional class weight to amplify the contribution to the loss for classes that are less abundant than others. When we considered the viral taxonomic realm for CV groups (see "CV and hyperparameter tuning"), we used the viral realm to compute class frequency labels for each virus. Specifically, we compute $C_i$ as an inverse abundance frequency. Suppose that the $i$th genome belongs to viral realm $k$, then the class weight is computed as:

$$C_i = \frac{N}{n_k} \quad (8)$$

where $n_k$ is the number of genomes in the training dataset belonging to viral realm $k$ out of $N$ total genomes. When the taxonomy of the viruses was not considered, such as when using protein diversity for CV (see "CV and hyperparameter tuning"), $C_i$ was equal to a 1-vector, meaning that each virus contributed equally to the loss. These PST models trained with TL are referred to as "PST-TL".

We implemented an easy positive semi-hard negative triplet mining strategy as described previously[15]. To account for the self-supervised choice of the negative genome, the scale factor $\omega_i$ reweights the anchor-negative distance according to the following exponential decay equation:

$$\omega_i = \exp\left( -\frac{\mathrm{CD}(G_i^a, G_i^n)}{2(c\sigma)^2} \right) \quad (9)$$

where $CD(X, Y)$ is the Chamfer distance between the genomes $X$ and $Y$, $c$ is a scaling factor, and $\sigma$ is the standard deviation of all Chamfer distances in a minibatch. $[\cdot]_+ = \max(0, \cdot)$, which means that there is no contribution to the loss function for cases where the positive genome is already closer to the anchor genome than the negative by a margin of $\alpha$. Thus, $\alpha$ is the farthest distance the negative genome needs to be from the anchor compared to the positive genome. This restraint notably prevents representation collapse that could occur in the naïve case of embedding the anchor and positive genomes in the same position.

For a training minibatch, positive mining occurs in the input ESM2 embedding space using Chamfer distance, before the PST forward pass and before concatenating positional and strand embeddings. Thus, the positive genomes are sampled online during training, and the candidates will differ each epoch as minibatches are formed. The Chamfer distance $CD(X, Y)$ between genomes $X$ and $Y$ always uses the input ESM2 embeddings and is defined as follows:

$$\mathrm{CD}(X, Y) = \frac{1}{|X|} \sum_{x \in X} \min_{y \in Y} ||x - y||_2^2 + \frac{1}{|Y|} \sum_{y \in Y} \min_{x \in X} ||x - y||_2^2 \quad (10)$$

where $x \in X$ are the proteins from genome $X$ and $y \in Y$ are the proteins from genome $Y$. Intuitively, this means that the positive genome is defined as the most similar genome based on cumulative distance of ESM2 protein embeddings, which should choose a positive genome that encodes the most similar proteins.

Negative mining requires the positive genome for a semi-hard sampling scenario, which is important for forming triplets that will lead to fast and efficient updates to the model parameters. Notably, negative mining also happens online and is affected by model updates because negative genomes are sampled in the learned PST embedding space. The only candidates for a negative genome are those that are farther than the positive genome in the PST embedding space using Euclidean distance, and we choose the first genome that is farther than the positive as the negative in the semi-hard case. In cases where there are no genomes farther than the positive genome in the PST embedding space, such as at the beginning of training when the model weights have not been well-optimized, we loosen the semi-hard sampling requirement and choose the genome closest to the positive genome as the negative genome. Since negative mining is self-supervised, we use the exponential decay reweighting factor $\omega_i$ to downweight poor choices of a negative genome that are actually very similar to the anchor genome. Notably, the $\omega_i$ reweighting factor depends on the Chamfer distance (Eq. 9) and, subsequently, the input ESM2 embeddings. Thus, we implicitly consider the ESM2 embeddings as a ground truth for protein representation when mining both the positive and negative genomes.

## PointSwap sampling

When training PST-TL models, we used the data augmentation technique PointSwap sampling[15]. During positive mining, we keep track of the most similar protein from the positive genome $X^p$ for each protein in the anchor genome $x \in X$ (Eq. 10) as the flow $x_i \rightarrow x_j^p$. We create the augmented genome $X'$ as follows:

$$X' = \mathrm{PointSwap}(X, X^p) = \left\{ x_0', \ldots, x_{n_i}' \right\} \text{ where } x_i' = \begin{cases} x_j^p & \text{if } u_i < p \\ x_i & \text{otherwise} \end{cases} \quad (11)$$

where $u_i$ is a set of samples from a standard uniform distribution $[0, 1]$ and $p$ is a rate of protein swapping between genomes. This means that the augmented genome $X'$ differs from the anchor genome by swapping related proteins with the most related positive genome, which intuitively mimics genetic variation. To form an augmented triplet with the augmented genome as the positive genome, the negative genome is selected from the set of augmented genomes in a minibatch using the procedure described above in "Training the PST foundation models with TL".

## A MLM PST variant

To compare different self-supervised objective functions, we implemented an MLM loss objective using only the PST encoder. A tunable fixed proportion of proteins are masked per genome, where masked protein embeddings are set to 0-vectors. We allow these 0-vector masked embeddings to be concatenated with the current positional and strand embeddings, rather than equally sized 0-vectors, since the position and strand are not unknown. In traditional large language models, only the word identity, and not position, is masked, since the contextual information of neighboring words is used to predict the identity of the masked word.

The PST encoder contextualizes protein embeddings according to the configuration of genome subgraphs (see "PST model architecture" and Eq. 3). We adapted a similar approach for genome language modeling with proteins represented as embeddings using MLM[10]. Briefly, we use a mean squared error function to mirror an MLM objective, since the protein words are $d$-dimensional vectors and not scalars:

$$\mathcal{L} = \frac{1}{n_m} \sum_{i=1}^{n_m} \left\| Y_{pred} - Y_{target} \right\|_2^2 \qquad (12)$$

where $n_m$ is the total number of masked proteins per minibatch, $Y_{target}$ is the target masked embeddings concatenated with positional and strand embeddings, and $Y_{pred}$ is the contextualized protein embeddings. We further considered that multiple proteins could occur in the same genomic contexts, such as homologs or reordering of grouped sets of proteins. To account for this, we positively sampled the nearest neighbor for each input protein embedding in ESM2 space using Euclidean distance and computed the PST-MLM protein embeddings, denoted as $Y_{pos}$. The mean squared error between $Y_{pos}$ and $Y_{target}$ was computed similarly and averaged with the $Y_{pred} - Y_{target}$ loss:

$$\mathcal{L} = \frac{1}{2n_m} \sum_{i=1}^{n_m} \left\| Y_{pred} - Y_{target} \right\|_2^2 + \left\| Y_{pos} - Y_{target} \right\|_2^2 \qquad (13)$$

Finally, to produce genome embeddings, which PST-MLM models do not directly produce, the contextualized protein embeddings are averaged over each genome.

## CV and hyperparameter tuning

To optimize the model hyperparameters (Supplementary Table 1), we used Optuna[57] (v3.3.0) to iteratively sample hyperparameters in a direction that optimizes the objective function using a Bayesian Tree-structured Parzen Estimator method. Model performance was evaluated using one of two different CV strategies each based on a LOGO approach.

The first CV strategy involved a modified LOGO implementation. Here, we considered the viral taxonomic realm to be the group with 5 total groups: Duplodnaviria, Monodnaviria, Riboviria, Varidnaviria, and Unknown/Other. We adapted the LOGO strategy to always include Duplodnaviria in each training fold, since this group of viruses accounted for 65.4% of the training dataset. This resulted in training 4 separate models validated on the remaining viral realms. This notably helped to ensure that the best model is chosen based on the relatively lower abundance viral taxa in the training dataset. Each of the four folds were synchronized during training to enable overall performance monitoring as the average of each fold. This enabled real-time monitoring of each tuning trial's performance. Thus, we were able to stop trials early depending on several criteria using the average validation loss of each fold: (1) if the loss plateaued (standard deviation of change less than 1e-6) after having trained three epochs, (2) if the loss did not decrease by 0.05 within five epochs, (3) if the current performance was worse than the median performance of previous trials at the same training epoch, (4) if the model was trained for 20 epochs, (5) if 24 h passed, (6) or if the loss was not finite. For number 3, this was tracked by the Optuna framework, and we required at least 1 complete trial before this was enabled. In the case of early stopping due to reasons 1, 2, 3, and 6, these trials were marked as pruned and not used by Optuna's median performance calculation. Models tuned with this taxonomic approach have "-T" in the model name (i.e., PST-TL-T_small).

The second LOGO CV strategy used groups based on protein diversity. We constructed five groups with relatively evenly distributed protein diversity using a greedy heuristic. First, we clustered the training proteins using mmseqs2 (v13.45111, cluster -s 7.5 -c 0.5

--cluster-mode 1) into remotely related groups of proteins. We then constructed a presence/absence binary matrix $V$, where the rows were each training virus genome and the columns were the remote protein clusters. We then computed the dice similarity scores $S$ between all pairs of viruses that shared any protein clusters to compute which viruses shared the same sets of proteins:

$$S_{ij} = \frac{2\left| v_i \cdot v_j \right|}{\left| v_i \right| + \left| v_j \right|} \qquad (14)$$

We then filtered $S$ to retain only genome-genome connections with a dice similarity ≥0.5, which resulted in 25,733 connected components of genomes that had similar protein content profiles. We computed the average dice score of all pairs of genomes within the same connected components and across connected components, which showed that there was not significantly shared protein content between connected components (Supplementary Fig. 4a). We then constructed four groups with minimally overlapping protein diversity using a greedy algorithm. First, the four largest connected components were taken as seeds for each of the four groups. To confirm that there was minimal overlap in protein diversity between these seeding connected components, we compared their average intra- and inter-connected-component dice similarity (Supplementary Fig. 4b). Then, for each of the four seeding connected components, we constructed four independent priority heaps based on the average dice similarity between genomes in each of the seeding components and genomes in all other connected components. Each of the four broad groups were expanded in round-robin fashion by popping off the most similar connected component from the heap and adding this to the current group if it had not already been taken. We further confirmed that there was greater internal similarity than inter-group similarity with an average maximum dice score, where the maximum non-self dice score was computed for each genome searching against all other genomes from each of the final 4 groups (Supplementary Fig. 4c). Finally, there were 1734 genomes that did not share any protein clusters with any other genomes and were, thus, not included in the above greedy search algorithms. Since these genomes do not share any protein similarity with any of the other genomes, we added all of these to the smallest group, since this would not affect the overlap in protein diversity of any groups. This resulted in four relatively balanced groups consisting of 31,778, 25,119, 24,650, 22,042 training genomes, respectively, with minimal overlap in shared protein content (Supplementary Fig. 4c). Models tuned with this protein diversity approach have "-P" in the model name (i.e., PST-TL-P_small).

The summary for the number of CV trials for each combination of training objectives (MLM vs TL), ESM2 input size (ESM_small vs ESM_large), and CV strategies can be found in Supplementary Table 3. Briefly, each of the 8 model setups had 7–45 complete trials, 5–78 trials that got pruned due to poor performance, and 1–16 failed trials. The only reason for failing was due to out-of-memory errors on A100 80GB vRAM GPUs hosted by the University of Madison-Wisconsin Center for High Throughput Computing[58]. All trials were tuned using 1 GPU since Optuna has limited support for GPU parallelism.

The final performance for each training trial was the average validation loss from each of the models trained in each CV fold. For PST-TL-T models, once the TL of the best model setup decreased below 20.0, we chose the best hyperparameter configuration, which can be found in Supplementary Table 2. All other models approached loss values below 1.0 without needing to arbitrarily stop. The summary of final training can be found in Supplementary Table 4, including the time needed to train on the full training set using 1 A100 GPU. The final models took between 4.5-33.7 h to train with early stopping once the training loss plateaued and did not decrease by 0.05 within five epochs. During training of the final models, a learning rate scheduler

was used that linearly decreased the learning each epoch. Further, we tested batch accumulation sizes of 1, 25, 50, 100, and 250, and the best model among these five accumulation sizes was chosen based on the minimum loss at the end of training. The selected batch accumulation size for each model is also reported in Supplementary Table 4.

For both tuning and training the final models, gradients were clipped to keep all values below a magnitude of 1.0, and we used mixed precision training, using bfloat-16 data when available. These choices helped stabilize training. Our fold training synchronization strategy and modified LOGO CV approach were implemented in a custom package called "lightning-cv" available from the main model repository. This package heavily relies upon and extends functionality in the lightning-fabric sub-library of PyTorch-Lightning (v2.0.7). For the protein diversity CV groups, we used the "LeaveOneGroupOut" implementation in scikit-learn (v1.4.2) that is integrated into "lightning-cv".

### GenSLM ORF and genome embeddings
We used the 25 M parameter GenSLM[30] foundation model ("genslm_25M_patric", downloaded September 2023) for our analyses since the output embedding dimension (512) was on par with other protein and genome embeddings used. The GenSLM foundation model is pretrained only on bacterial and archaeal nucleotide genes where the gene sequences were broken into codons as input. The authors then finetuned the foundation models on a dataset of SARS-CoV-2 genomes. However, it is not clear if only the ORFs from SARS-CoV-2 were included or if entire viral genomes were used as input during finetuning. This is further complicated by the fact that the protein-coding density of the SARS-CoV-2 genome is 71.2% (based on the NCBI RefSeq reference sequence NC_045512.2). We chose to mimic the pretraining setup and input the protein-coding ORFs for each virus in our datasets. Notably, we used GenSLM as a nucleotide analog of ESM2, producing ORF embeddings akin to the ESM2 protein embeddings. We used these ORF embeddings for protein/ORF analyses and the average of these over each genome as genome embeddings for genome analyses.

### HyenaDNA genome embeddings
We used the HyenaDNA[31] model with the longest context size (1 M nucleotides, "large-1m", downloaded from HuggingFace in November 2023) that has 6.5 M parameters. We converted all non-ACGTN nucleotides to Ns. Genomes larger than 1 M nucleotides were split into non-overlapping fragments of 1 M nucleotides at most. Then each fragment was tokenized and fed to the "large-1m" HyenaDNA model. The embedding of each genomic fragment was averaged to produce the final genome embedding. We also used this same averaging approach for fragmented genomes (i.e., vMAGs), where the final genome embedding was the average of each fragment.

### Tetranucleotide frequency vectors as genome embeddings
For each genome, we computed tetranucleotide frequency vectors ($\begin{bmatrix} AAAA & \cdots & TTTT \end{bmatrix}$) using the bionumpy[59] package (v1.0.8). We filtered all nucleotides not in the canonical ACGT alphabet before calculation. For RNA viruses, U nucleotides were represented by T for simplicity. For multi-scaffold viruses, these frequency vectors were computed for each scaffold and then averaged over each scaffold. Throughout the paper, these are referred to as "kmer".

### Clustering genome and protein embeddings
We constructed a similarity-weighted kNN graph. The set of kNN was computed using the faiss[60] (v1.8.0) Python bindings. We used the divide-and-conquer IndexIVFFlat search index that splits the input embeddings into $n_{cells}$ Voronoi cells for faster retrieval. The number of Voronoi cells was $n_{cells} = 3878$ for clustering IMG/VR v4 genomes, $n_{cells} = 4096$ for clustering IMG/VR v4 proteins, $n_{cells} = 329$ for clustering MGnify genomes, and $n_{cells} = 2048$ for clustering MGnify proteins.

Genome or protein clusters were then detected in the similarity-weighted kNN graph using the Leiden[34] algorithm Python implementation of iGraph (v0.11.3) with a resolution of 0.9 to only cluster embeddings with a high angular similarity. Lowering the similarity threshold to 0.6, which is a very relaxed threshold as this value indicates nearly perpendicular vectors, did not substantially change the number of clusters or average cluster size for any embedding. This likely reflected the vast diversity of the viral genomes and proteins, so we opted to keep a high embedding similarity threshold of 0.9 for clustering. For each clustering iteration, the random number generator was reset and seeded using the same random seed, so all clustering results are exactly reproducible. We do not include singletons as clusters for downstream analyses.

### Genome and protein clustering purity
To compute the overall purity of clusters produced by different genome and protein embeddings, we computed viral and host taxonomic purity for each set of genome clusters and functional purity for each set of protein clusters using the information gain ratio $I$ as a proxy for purity. For taxonomic purity, we considered the case of clustering all genomes into a single cluster as the background. For functional purity using curated functional categories from VOG or PHROG, we used the distribution of functional categories from the annotation databases as the background. In either case, unlabeled proteins and genomes were excluded during the entropy computation but included for the cluster size weighting. Then we computed $I$ as follows:

$$I = \frac{H_{background} - \sum_{i=1}^{n_{clusters}} H_i w_i}{H_{background}} \in \begin{bmatrix} -h, 1 \end{bmatrix} \tag{15}$$

where $h$ is the information gain of the background compared to a uniform distribution and H is the information entropy of each cluster with respect to a set of labels related to viral taxonomy, host taxonomy, or protein function. The calculation of I is weighted by the size of each cluster, including unlabeled members, where:

$$w_i = \frac{n_i}{\sum_{i=1}^{n_{clusters}} n_i} \tag{16}$$

Values of $I$ close to 0 indicate clustering patterns with no improvement above background, while values of $I$ close to 1 suggest maximal purity since there are few clusters with multiple labels. It is possible to interpret $I$ as a purity score since the backgrounds are not pure with respect to the labels. Thus, a maximum $I$ means that there is only a single label for each cluster.

### Average amino acid identity (AAI)
We used mmseqs2[61] (v13.45111) and polars (v1.24) to compute the AAI between pairs of viruses at scale. Given the large number of viruses in this study (>250 k), we did not exhaustively compute the AAI between all pairs of viruses (~ 32.5B). Instead, we used heuristics implemented by mmseqs2 to only consider the AAI between viruses that had any protein sequence similarity detectable when using the mmseqs2 search settings: -s 7.5 -c 0.3 -e 1e-3. For each pair of viral genomes, we only retained the best hit for each protein from each genome. Then, AAI was computed as the mean of protein-protein sequence similarities computed by mmseqs2. Since we did not exhaustively compute all possible protein-protein sequence alignments, some AAI scores were only based on small fractions of the total number of proteins encoded in each genome. To account for this, we took the harmonic mean of AAI and the minimum proportion of shared genes to use as a final AAI-based score when necessary.

## Average structural identity (ASI)

We compiled foldseek[36] (v284d732b9a801c642213a396286-cae3344a11d2c) from source to enable CUDA support. We used foldseek to download the ProstT5 model to translate protein sequences to 3Di structural sequences for all proteins in both test datasets using A100 GPUs. We split all datasets into subsets of 100,000 proteins per file to parallelize this translation on a computing cluster and then concatenated the output files into a single foldseek database. We then computed ASI analogously to AAI, except using foldseek to search our 3Di structure sequence databases with the following settings: -c 0.3 -e 1e-2 -s 4.0. The 3Di searches were performed on CPU.

## Protein functional annotation

We used VOG (r219) and PHROG[62] (v4) databases for the annotation of viral proteins. For VOG, which supplies profile Hidden Markov models (HMMs), we used pyhmmer (v0.9.0) with a bit score cutoff of 40. For PHROG, we used mmseqs2 (v13.45111) with the recommended search settings (https://phrogs.lmge.uca.fr/READMORE.php). In both cases, we kept the best hit for each protein with the max bit score. For each database, we curated the functional categories of each annotation that we describe below.

For PHROG, which already provides an extensive set of ten categories (including unknown function), we manually readjusted certain categories. Our manual curation of the PHROG database affected 1937 out of 38,880 profiles. We renamed the following categories for better intuition of the functional category: "DNA, RNA and nucleotide metabolism" to "nucleotide metabolism", "integration and excision" to "lysogeny", and "transcription regulation" to "gene expression". We then dissolved the "moron, auxiliary metabolic gene and host takeover" category for being too broad and relatively unrelated. These 461 profiles were split into the already existing "other"; the newly created "host takeover", "lysogenic conversion", "metabolic gene"; and the renamed "gene expression", "lysogeny", and "nucleotide metabolism" categories. Generic annotations like "membrane associated protein" and "ABC transporter" were put in the "other" category. We considered proteins involved in host replication and cell division inhibition, superinfection exclusion, anti-sigma factors, and defense against host antiviral proteins to be "host takeover". Proteins that encoded toxins or antitoxins/resistance proteins were categorized as "lysogenic conversion." Proteins directly involved in specific metabolic transformations were put in "metabolic gene," while accessory or generic proteins like "nicotinamide mononucleotide transporter" were considered as "other". These changes can be found in Supplementary Data 3.

VOG provides very broad categories: "Xr" for replication, "Xs" for structural, "Xh" for host-benefitting, "Xp" for virus-benefitting, and "Xu" for hypothetical proteins. The "Xh" and "Xp" categories are also ambiguous on what specific function the protein may perform. We, therefore, used text pattern matching on the specific HMM annotation descriptions to subdivide all HMMs into 9 categories: anti-host defense, exit, gene expression, integration, packaging, replication, structural, other, and unknown. Briefly, we separated terminases, portal proteins, and head packaging proteins from other structural proteins into a "packaging" category. Lysis, virion export, and budding HMMs were considered collectively as the "exit" group. "Integration" includes both integrases and excisionases as well as transposases. We considered all nucleotide metabolism and genome replication to be part of "replication". To account for overlap in text matching, we enforced the following hierarchy: structural > packaging > exit > integration > gene expression > anti-host defense > replication > unknown > "RNA polymerases" > other. The final category for each HMM was, therefore, the highest in the hierarchy. We added RNA polymerases that did not indicate if they were replicative RNA-directed or transcriptive DNA-directed at the bottom to put these specific RNA polymerases in the "gene expression" category. Additionally, HMMs without matches were thus considered in the "other" category. The

category for each VOG r219 HMM can be found in Supplementary Data 4, and the regex patterns used to categorize each HMM can be found in Supplementary Table 5.

## Protein attention scaling and analysis

We computed the attention values as follows: Let $A_{ij} \in A_i$ be the mean attention score across all attention heads for the $j$th protein from the $i$th genome:

$$A_{ij} = \frac{1}{n_{\text{heads}}} \sum_{k=1}^{n_{\text{heads}}} A_{ijk} \tag{17}$$

The sum of per-protein attention values for each genome is 1.0:

$$\sum_{A_{ij} \in A_i} A_{ij} = 1.0 \tag{18}$$

Given $n_i$, the number of proteins in the $i$th genome, and $n_k$, the number of proteins in the $k$th genome, and $n_i \neq n_k$, it follows that $A_i$ and $A_k$ are not directly comparable since the number of proteins each genome is not the same. More explicitly stated, consider $n_i = 2$ and $n_k = 4$, and $A_i = \begin{bmatrix} 0.5 & 0.5 \end{bmatrix}$ and $A_k = \begin{bmatrix} 0.5 & 0.3 & 0.05 & 0.15 \end{bmatrix}$. For the genome $i$, the model has uniformly split attention to both proteins, while for genome $k$, the model clearly has attended to the first protein more than the others, despite the numerical values being equivalent.

Therefore, to compare the PST-TL attention values per protein for each scaffold, we normalized the attention scores. We considered the background case for the attention distribution $A_i$ to be a uniform distribution, ie $A_i \sim U(0,1;n_i)$ where $U(0,1;n_i)$ is a standard uniform distribution with probability $\frac{1}{n_i}$ of attending any of the $n_i$ proteins. We then computed the distance between $A_i$ and $U(0,1;n_i)$ using the normalized Kullbach-Leibler (KL) divergence:

$$D_i = \frac{H_i^{U(0,1;n_i)} - H_i^{A_i}}{H_i^{U(0,1;n_i)}} \in [0, 1] \tag{19}$$

$H(X)$ is the entropy of the probability distribution $X$:

$$H(X) = -\sum_{x \in X} p(x) \log_2 p(x) \tag{20}$$

We then rescale all per-protein attention values in $A_i$ by the KL-divergence $D_i$ to down-weight misleadingly large attention values that are uniformly (randomly) distributed:

$$A'_{ij} = A_{ij} \times D_i \tag{21}$$

Thus, for cross-genome comparisons, we use the normalized attention scores $A'_i = \left\{ A'_{i1}, \ldots, A'_{in_i} \right\}$. In our above example, $A'_i = \begin{bmatrix} 0.0 & 0.0 \end{bmatrix}$ and $A'_k = \begin{bmatrix} 0.0881 & 0.0528 & 0.0088 & 0.0264 \end{bmatrix}$.

To associate PST-TL attention with protein function, we first chose the top 1000 proteins for each VOG and PHROG category based on the rescaled attention values $A'$. We normalized the values from the PST-TL-T models and PST-TL-P models by dividing by the maximum scaled attention for each model. This allowed us to focus on relative differences in attention, rather than compare absolute differences between these two sets of models. To compare attention values with abundance in the training and test datasets, we then clustered the proteins using sequence identity (mmseqs2 v13.45111 -e 1e-3 -c 0.5 -s 7.5). We computed the max scaled attention $A'_{ij}$ for all proteins in the same cluster, excluding singletons.

## Protein annotation improvement

We considered all proteins unable to be annotated using the VOG r219 databases as hypothetical proteins, where $N_H$ is the number of hypothetical proteins. $N_H$ includes both proteins detectable by VOG profiles but assigned a hypothetical function and proteins undetectable entirely. We computed the annotation improvement as a function of a protein/ORF embedding. We used the same protein search settings described in the Methods section "Clustering genome and protein embeddings": angular similarity on the unit-normalized protein embeddings. For each protein, we searched for $k$ nearest neighbors, scoring this as an improvement if any neighbor proteins in that set were annotated:

$$T_k = \sum_{i=1}^{N_H} \begin{cases} \text{if ANY of the } k \text{ Nearest Neighbors of } i \text{ is annotated} & 1 \\ \text{else} & 0 \end{cases} \quad (22)$$

Then, we computed the overall annotation improvement $AP_k$ as the proportion of hypothetical proteins whose nearest neighbors had any annotations:

$$AP_k = \frac{T_k}{N_H} \quad (23)$$

To enforce a greater degree of confidence beyond any neighbors being annotated, we further required that all $k$ nearest neighbors belong to the same VOG functional category. For example, this functional category set of two neighbors {structural, replication} would not be counted as an annotation improvement, since the two nearest neighbors have broadly different functions. We excluded nearest neighbors that were unannotated from this penalty (i.e., {unknown, structural, unknown, structural} counts as a hit). The updated scoring function can be described as:

$$T_k = \sum_{i=1}^{N_H} \begin{cases} \text{if ALL of the } k \text{ Nearest Neighbors of } i \text{ have the same VOG category} & 1 \\ \text{else} & 0 \end{cases} \quad (24)$$

## Protein function co-clustering

We used curated PHROG functional categories (Supplementary Data 3) to compute functional co-clustering, excluding the category for proteins of unknown function. Given a protein clustering configuration, for each protein cluster $P_i \in P$, we count the co-occurrence between pairs of functional categories $u$ and $v$ as the product of the number of proteins belonging to each category in the cluster:

$$C_i^{uv} = n_i^u \times n_i^v \quad (25)$$

where $n_i^u$ is the number of proteins in the $i$th protein cluster that belongs to category $u$. The observed co-occurrence $C^{uv}$ between the functional categories $u$ and $v$ is defined as the sum of cluster-level co-occurrences:

$$C^{uv} = \sum_{i=1}^{|P|} C_i^{uv} \quad (26)$$

To account for the distribution of PHROG annotation profiles, we computed an enrichment score against the background of the distribution of the 38,800 PHROG profiles:

$$E^{uv} = \frac{C^{uv}}{C_{background}^{uv}} \in [0, \infty] \quad (27)$$

where $C_{background}^{uv}$ is computed analogously to $C^{uv}$ except using relative abundances of the annotation profiles themselves instead of annotated proteins. To identify functional categories that co-occur frequently, we constructed a fully-connected graph with all PHROG functional categories as nodes and the corresponding edge weights $E^{uv}$ between categories $u$ and $v$. We then applied the Leiden algorithm with resolution 1.25 to identify sub-communities of co-occurring functions enriched above background.

## Protein functional module detection

We defined the following protein functional modules based on curated functional categories (Supplementary Data 3,4) and annotation text searches. For PHROG DNA-interacting modules, we included all hits that belonged to either "nucleotide metabolism", "lysogeny", or "gene expression" categories. For VOG, all hits belonging to "replication", "integration", "packaging", and "gene expression" were included. For PHROG late genes, annotations in the categories "tail", "head and packaging", "connector", and "lysis" were retained. Likewise, for VOG, the categories "structural", "exit", and "packaging" were included.

We considered protein clusters to correspond to a specific functional module if they met the following module-specific criteria: (1) We required at least two categories to be represented by the annotated proteins in each cluster. (2) We excluded protein clusters that had any annotated proteins outside the indicated functional categories to focus on protein clusters that most strongly fit our definition of functional modules, i.e. were "pure" relative to the definition of the functional module. We ignored proteins of unknown function when considering functional purity.

## Capsid structure searches

To quantify the frequency at which embedding-based protein clusters co-cluster VOG-detectable capsid proteins with proteins unable to be assigned function by VOG, we excluded all embedding-based protein clusters that did not solely consist of annotated capsids and hypothetical proteins. This led to a total of 462,141 IMG/VR v4 test proteins and 7448 MGnify test proteins for this analysis.

We used foldseek[36] (v284d732b9a801c642213a396286-cae3344a11d2c) to convert our protein sequence database into a 3Di-structure database using the ProstT5[37] model (downloaded February 2025; foldseek createdb with "--prostt5-model" option; see "Average Structural Identity (ASI)"), which uses language tokens to represent structural features. We searched our 3Di-structure database against 295k structures from the Protein Data Bank[38] (PDB; downloaded using foldseek in July 2024) using default settings. We selected only alignments with bit scores ≥100 and alignment identities ≥10%. To select PDB structures corresponding to capsid proteins, we chose only PDB structures from the foldseek PDB database whose description contained either "capsid" or "virion". We validated this approach by aligning AlphaFold 3-modeled[40] (https://alphafoldserver.com) monomer structures with the HK97 major capsid protein (2FS3) using TM-align[63] implemented in the PDB web server[64]. We choose 2 proteins with the highest scoring structural alignment as determined by foldseek, each from either proteins annotated with a VOG profile of unknown function or proteins undetected by VOG, for this analysis.

We then scored the proportion of unannotated proteins from each cluster that had a structural alignment with a PDB capsid protein out of all the searched proteins from each test set.

## Embedding UMAP visualization

We used the Python implementation of the UMAP algorithm[65] (umap-learn v0.5.3) for embedding visualization only. For genome embeddings, we used the default 15 nearest neighbors defined using Euclidean distance. When computing the reduced embeddings, we jointly embed the genome embeddings of both the training and test datasets for each type of genome embedding into the same space. We did not reduce the dimensionality before visualization, so the

embeddings themselves were directly used as inputs to the UMAP algorithm.

## Graph-based host prediction framework

For the virus-host prediction proof-of-concept, we modeled our framework off CHERRY[41], which applies graph learning on a virus-host interaction network $G = (X, E)$. Our implementation uses PyTorch (v2.2.2), PyTorch-Geometric (v2.5.2) and PyTorch-Lightning (v2.2.4). The interaction network is bipartite, meaning that there are two types of nodes: viral nodes $v_i \in V$ and host nodes $h_i \in H$. The total node set $X$ is thus $X = V \cup H$. The edges $E$ represent known virus-host pairs and may also include confident virus-host predictions that come from virus-host genome alignments (see "Host prediction training and test datasets" for more detail). Given $G$, the objective is a link prediction task to infer for any virus-host pair $(v_i, h_i)$ the probability of an edge existing in the interaction network.

For the most comparable analyses, we designed our neural network architecture based on CHERRY: an encoder consisting of multiple Graph Convolution[66] (GCN) layers and a decoder that performs the link prediction. The encoder propagates information in the genome embeddings among local neighborhoods. Specifically, the GCN encoder layers $e^{(l+1)}$ can mathematically be represented as:

$$e^{(l+1)} = \phi\left(\widetilde{D}^{-\frac{1}{2}}\widetilde{A}\widetilde{D}^{-\frac{1}{2}}e^{(l)}W^{(l)}\right) \quad (28)$$

where $l$ is the layer index, $\widetilde{A}$ is the adjacency matrix with self-connections ($\widetilde{A} = A + I$ and $I$ is the identity matrix), and $\widetilde{D}$ is the diagonal matrix where $\widetilde{D}_{ii} = \sum_j \widetilde{A}_{ij}$. $\phi$ is the activation function, and $W^{(l)}$ is the $l$-layer model weights. $e^{(0)} \in \mathbb{R}^{|X| \times N}$ is the input genome embedding where $N$ is the size of the embedding. To compare our work to CHERRY, which uses tetranucleotide frequency genome embeddings, we substitute the genome embedding with either the PST genome embeddings or the simple average of the ESM2 protein embeddings over each genome for both the viruses and hosts.

The decoder is a 2-layer feedforward neural network that takes the outputs from the encoder as input. We consider all possible virus-host pairs $(v_i, h_j) \in X$ as a query set $Q$ where each pair is represented by the difference in encoder embedding:

$$q_{ij} = \text{encoder}(v_i) - \text{encoder}(h_j) \quad (29)$$

The decoder, therefore, is mathematically written as:

$$\begin{cases} q_{ij}^{(l+1)} = \phi\left(q_{ij}^{(l)}\theta^{(l)}\right) \\ \text{decoder}(q_{ij}) = \text{sigmoid}\left(q_{ij}^{L-1}\right) \end{cases} \quad (30)$$

where $q_{ij}^{(l)}$ is the hidden feature in the $l$th layer out of $L$ total decoder layers, and $q_{ij}^{(0)} = q_{ij}$. $\phi$ is the activation function, and $\theta^{(l)}$ represents the weights of the $l$th fully connected layer.

## Host prediction training and test datasets

For the virus-host prediction proof-of-concept, we modeled our framework off CHERRY[41], which applies graph learning on a virus-host interaction network. To construct the network of known virus-host pairs, we used the train and test datasets from iPHoP[42]. Specifically, the train dataset included 3628 complete bacterial and archaeal viruses from NCBI RefSeq prior to 2021. The iPHoP test dataset consisted of 1636 complete bacterial and archaeal viruses from NCBI GenBank, distinct from the training dataset. Although both datasets indicate the taxonomy of the host, they do not provide specific genome accessions to link the viruses, which are necessary to construct the interaction network.

For the training dataset, we used the Virus-Host Database[67] (accessed April 2024) to determine the full host taxonomy. We then selected either the NCBI RefSeq representative sequence associated with the host taxonomy, if one existed, or the most complete (longest and assembly_level == "Complete Genome") genome from NCBI GenBank (accessed May 2024). We included all hosts in the Virus-Host Database if there were multiple, such as in the case of viruses with a relatively broad host range. The set of hosts notably includes multiple strains of the same species or species of the same genus as indicated in the Virus-Host Database. Then, any strain information was ignored, so the lowest level of evaluation was at the host species.

We performed a similar search for the test dataset using the information provided in Supplementary Table 2 of iPHoP[42]. We divided the test virus hosts whose species ranks were unknown (i.e., "Wolbachia sp.") into two different sets. If these hosts were already in the set of hosts for the training dataset, we did not retrieve any new host genomes. Instead, we considered all hosts currently in the set of hosts with the same genus as potential hosts for these viruses. For new hosts, we used the same search criteria as above to add a single new host for each of these viruses. This resulted in a total of 805 host genomes, corresponding to 594 unique host species. Supplementary Data 5 lists all viruses used by the iPHoP training and test datasets and their hosts.

## Constructing the virus-host interaction network

To construct the virus-host interaction network, we constructed the heterogeneous graph $G$ that has two node types (virus, host) and two edge types (virus-related to-virus, virus-infects-host). For the virus-host edges, we included all virus-host pairs identified above, meaning that $G$ includes both training and test viruses. We notably deviated from the CHERRY implementation by excluding confident host predictions that would have come from virus-host BLASTn genome alignments (proviruses) or CRISPR spacers. This deviation is not concerning since we focused on the relative performance of our PST genome embeddings compared to other tools and genome embeddings, rather than absolute predictive ability.

To select virus-virus edges representing pairs of similar viruses, we used a protein sharing network clustering approach when using tetranucleotide frequency genome vectors, since this is the same approach CHERRY uses[41]. We first excluded all singleton proteins since these do not inform about genome-genome relatedness and only serve to account for the proportion of gene sharing relative to the total number of proteins/protein clusters in each genome. After protein clustering using mmseqs2 (v13.45111 -s 7.5 -e 1e-3 -c 0.5) and filtering singleton proteins, we constructed a sparse $(|V| \times n_{pc})$ presence-absence matrix where $n_{pc}$ is the total number of protein clusters in the dataset. Each row represents what protein clusters are encoded in the indicated genome. We then computed the dice similarity $S_{ij}$ for each pair of genomes using Eq. 14. We then constructed a graph with all viruses where the edges are $S_{ij} | S_{ij} > 0$. To understand which viruses were considered related, we clustered this graph with the Leiden algorithm with a resolution of 0.1. Edges were created in the interaction graph between all viruses in the same gene-sharing clusters. For the other genome embeddings we tested, we considered pairs of viruses to be related if their genome embeddings were at least 75% similar based on angular similarity of unit-normalized embeddings. This ensured that each virus had at least 1 neighbor and was, therefore, included in the interaction graph. We then pruned these embedding-based virus-virus connections to maintain up to the top 15 neighboring viruses for each virus.

## Host prediction model training

We trained new graph-based host prediction models using the iPHoP training dataset, swapping the genome representations for PST genome embeddings or the simple average of the ESM2 protein embeddings

over each genome. We used a binary cross entropy loss objective for the link prediction task to classify the edge $E_{ij}$ as existing (1) or not (0):

$$\mathcal{L} = -\frac{1}{N}\sum_{k=1}^{N} y_k \log(p(y_k)) + (1 - y_k)\log(1 - p(y_k)) \qquad (31)$$

where $y_k$ is the discretized final output for the $k$th virus-host pair from the model decoder, given a probability threshold for whether an edge $E_{ij}$ is predicted to exist.

During training, we randomly split all virus-host edges $E = \{E^{(T)}, E^{(V)}\}$ into disjoint training $E^{(T)}$ and validation $E^{(V)}$ sets at an 80:20 ratio. We additionally randomly sampled negative edges $E' = \{E'^{(T)}, E'^{(V)}\}$ that do not exist in the virus-host interaction network $G$ to provide the model with negative examples (implemented by PyTorch-Geometric). The negative edge sets $E'^{(T)}$ and $E'^{(V)}$ are also disjoint, and $|E| = |E'|$ so that there was not label imbalance. During the message-passing stage of the model encoder, only the real edges $E$ are used. After message passing updates the node representations, we used $E \cup E'$ for decoding and inference with both real and negative edges. Therefore, $N = |E^{(\cdot)} \cup E'^{(\cdot)}|$ for either the training or validation edges. Since the prediction task does not depend on virus-virus edges, these edges were not split or negatively sampled. This means that the graph structure and message passing consider all viruses, not just training viruses. Thus, during training, we masked any virus-host edges that contain test viruses in the loss computation to prevent data leakage.

Although we strived to implement a nearly 1:1 model with the original CHERRY implementation, our implementation and training deviates in three ways. (1) We allowed separate learnable weights for each type of edge (virus-virus, virus-host, and host-virus) in the message-passing encoder layers by updating Eq. 28:

$$W^{(l)} = \begin{cases} W_{vv}^{(l)} & \text{virus} - \text{virus edges} \\ W_{vh}^{(l)} & \text{virus} - \text{host edges} \\ W_{hv}^{(l)} & \text{host} - \text{virus edges} \end{cases} \qquad (32)$$

$W_{hv}^{(l)}$ and subsequently host-virus edges are required to ensure reciprocal message passing between virus and host nodes despite the intuitive way of representing virus-host edges as directed. The native CHERRY implementation does not allow for edge type-specific weights, instead sharing weights for all edge types.

(2) Due to modeling $G$ as a heterogeneous graph, the message passing layer is not a true GCN layer, which is not implemented for heterogeneous graphs in PyTorch-Geometric. Specifically, we use a generalization of the GCN layer[68] that allows for heterogeneous graph learning with multiple node and edge types. For this layer, however, the node update equations for this layer and the GCN layer are identical, but there may be PyTorch-Geometric implementation-specific differences beyond changing node representations.

(3) We explored a more sophisticated technique for handling the training and validation splits for link-level tasks that we refer to as "disjoint training". Specifically, we divided the real training edges $E^{(T)}$ into 2 disjoint sets $E^{(T)} = \{E^{(MP)}, E^{(D)}\}$ where $E^{(MP)}$ are edges only used for message passing (node updates) and $E^{(D)} \cup E'^{(T)}$ are edges only used for supervision (decoding and inference). Specifically, $E^{(D)} \cup E'^{(T)}$ are the edges used for link prediction. We only considered a 70:30 split for $E^{(MP)}$ and $E^{(D)}$ for this study when this was enabled. This modification is analogous to splitting training data into separate training and validation sets to prevent data leakage but only for training edges.

To decouple the effect of these three differences from the choice of node embeddings, we trained a model that is nearly faithful to the CHERRY implementation without these changes (barring the required

change #2), and then we trained a separate model using tetranucleotide frequency genome embeddings (kmer) that enables our changes. Thus, the CHERRY and "kmer" model use the same virus-host interaction graph as input, but the "kmer" models explored the effects of changes #1 and #3.

To lightly optimize hyperparameters, we sampled from sets of intuitive values for the number of encoder layers, decoder hidden dimensions, learning rate, whether to enable disjoint training (at a 70:30 split), and whether to allow edge specific-weights in the encoder or not. We did not dilate the input embedding dimension in the encoder layers. For the 2-layer feedforward decoder network, we only chose values smaller or equal to the input embedding dimension for the first layer. The second layer dimensions were then required to be strictly less than the first layer dimensions. See Supplementary Table 6 for the values sampled for each hyperparameter. We applied the same random seed when training each iteration and chose the best model based good overall performance and lowest validation loss at the end of 150 training epochs. We defined "good" overall performance as a validation loss curve that was monotonically decreasing over or constant at the end of training time. We selected a total of seven models that were the best: CHERRY without the above changes and 6 that allowed the above implementation changes and used different genome embeddings. All models were trained with a dropout of 0.25 after the encoder and after each decoder feedforward layer. We used the ReLU activation function after each layer.

## Host prediction model evaluation

iPHoP (v1.3.3) and the models trained in this study were evaluated using the iPHoP test dataset (see "Host prediction training and test datasets"). For the graph-based models, we considered all test virus-host pairs for link prediction and retained only those ≥75% confidence, which is the minimum for iPHoP, or ≥90% confidence. All virus-host pairs were considered to enable resolution at each host taxonomic rank. However, we only evaluated if the true host taxon was among the predictions above the confidence threshold, so not all predictions were analyzed. Specifically, we computed the proportion of the iPHoP test viruses whose true host taxon was confidently predicted among all hosts.

Since there were notably a nontrivial number of viruses in the iPHoP test dataset that were similar to those in the PST training dataset based on AAI (see "Average amino acid identity (AAI)"), we filtered these viruses out using several similarity cutoffs to evaluate their effects on our interpretation of the host prediction results. More specifically, the similarity of each iPHoP test virus to the closest PST training virus was computed as the harmonic mean of the AAI and the minimum proportion of genes used to compute AAI relative to each genome.

## Statistics & reproducibility

No statistical hypothesis tests were performed in this study. For analyses in this study, generally no data were excluded except as indicated in a relevant section above (i.e., excluding proteins of unknown function when scoring a function-based metric). Statistical summary metrics like Pearson correlation were computed without considering significance from a hypothesis test, since the purpose was comparative only. All analyses were performed using Python and various packages as indicated in a relevant section above. Pearson correlation was specifically computed with SciPy (v1.13.0) using the pearsonr function.

## Reporting summary

Further information on research design is available in the Nature Portfolio Reporting Summary linked to this article.

## Data availability

Sources for publicly available viral genomes are listed in Supplementary Data 1. All data specific to this manuscript, including protein FASTA files, protein and genome embeddings, trained PST model weights, virus-host interaction graphs, Supplementary Tables, Supplementary Data, and Source Data used for figure making, were deposited at DRYAD: (https://doi.org/10.5061/dryad.d7wm37q8w). Data can be downloaded via the DRYAD web interface, and we have also provided command line access to download files with instructions in the PST GitHub repository. Descriptions of the Supplementary Data can be found in the Supplementary Information file, and all other necessary information is in the DRYAD README. The protein embeddings from PST-MLM models are not provided due to storage limitations in the DRYAD repository.

## Code availability

All code for the PST model architecture and analyses specific to this manuscript is available at https://github.com/AnantharamanLab/protein_set_transformer. We have indicated the specific commits upon manuscript submission and resubmission for maximal reproducibility, but we note that all code changes since then have not affected the main model components. Specifically for manuscript-associated analyses, Jupyter notebooks are provided for each method section that uses code. We provided additional repositories for generating the ESM2 protein embeddings, GenSLM ORF and genome embeddings, and HyenaDNA genome embeddings that can be found in the main model repository above.

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

## Acknowledgements

This research was supported by a National Institute of General Medical Sciences of the National Institutes of Health award (R35GM143024, KA) and by a National Science Foundation award (2226451, AG). CM was supported by a National Science Foundation Graduate Research Fellowship and a University of Wisconsin-Madison SciMed GRS Fellowship. AG acknowledges support from Jeanne M. Rowe. We thank members of the Anantharaman lab for project discussion and feedback on the manuscript. Model training and inference was performed using the resources and assistance of the University of Wisconsin-Madison Center for High Throughput Computing.[58] We thank Drs. Yunha Hwang, Jeffrey Ruffolo, and Mark Craven, experts in biological machine learning and data science, for their assessment of potential biosecurity risks of this study.

## Author contributions

All authors (C.M., A.G., and K.A.) conceived the project. C.M. conducted model and software development, all analyses, results visualization, and content organization. A.G. and K.A. provided project feedback. C.M. and K.A. wrote the manuscript draft. All authors (C.M., A.G., and K.A.) reviewed the results and edited and approved the manuscript.

## Competing interests

The authors declare no competing interests.
