## [Transparent Peer Review file · Nature Communications]

Protein Set Transformer: A protein-based genome language model to power high diversity viromics

Corresponding Author: Dr Karthik Anantharaman

Version 0:

Reviewer comments:

Reviewer #1

(Remarks to the Author)

PROTEIN SET TRANSFORMER: A PROTEIN-BASED GENOME LANGUAGE MODEL TO POWER HIGH-DIVERSITY VIROMICS.

Cody Martin, Anthony Gitter, and Karthik Anantharaman

The paper by Martin, Gitter, and Anantharaman describes the creation of viral-specific protein language models and their use in various challenges holding back viral bioinformatics, including gene identification, protein annotation, and viral host prediction. Overall, the paper is well-written, grammatically well-formed, excellent and a novel contribution to the field, and I support its publication.

The pLMs described in this paper start with the ESM2 framework, and they have layered on additional, genome-specific attributes, notably the strand and location of the the genes that encode each protein. The novelty is enhanced in training the language models by introducing triplet training to provide both positive and negative mining of closely related genomes. This approach allows the authors to establish point swapping in the architecture to simulate evolution during data augmentation and then use the augmented data in the loss function.

The authors tuned two models—a small model with six layers and 8 million parameters and a larger model with 30 layers and 150 million parameters, and the training took proportionally longer for the large model, as expected.

They applied the learned Protein Set Transformer models to a testing set of viral genomes and demonstrated that the model learns “meaningful” information about the genome. Figure 2 demonstrates that new information has been learned and that the protein methods generate more meaningful information than the DNA-based methods.

I was excited to see that the results here recapitulate other results demonstrating that protein-based methods are far superior to nucleotide methods when clustering viral genomes. I also appreciated the limitations the authors highlighted in lines 256-261, comparing their results to existing taxonomy and host predictions.

I found several figures, especially Figs. 2-4, to be overly complex and confusing because they contain too much information, much of which is unnecessary. For example, there is no description in the Fig 2 legend about the clustering resolution, and I am not sure it adds much to the story because it is yet another parameter that can be varied.

Line 250 says, “Notably, pst-small genome clusters have the highest AAI among all methods (Fig. 2E)”, but it also has more, smaller clusters, so just the highest AAI is not that notable.

Next, the authors tackled the thorny subject of viral protein annotations, and their data in Fig. 3 demonstrates well-defined protein clustering with related functions.

They have clearly demonstrated the benefits they have brought to the data by showing that ESM2 embeddings did not lead to interpretable functional modules (lines 349-351), whereas their vPST, including genome location, adds functionality. Again, I think Fig. 3 could be simplified for clarity. For example, Fig 3 D PHROG panels show no difference, and perhaps Fig

3C could be simplified to fewer examples. I am not sure the figure will be improved, but the scaled attention in Fig. 3A should be the same scale for PHROG or VOG. Alternatively, you might consider eliminating PHROG or VOG data from the entire figure.

Fig. 4 could also be simplified by removing the genome clustering resolution – especially as that is secondary to the story. In panel C, the authors used low and med resolution, but in panel D, they used med and high resolution, which provides the perception of cherry-picked data.

Finally, the authors tackle the challenge of host-species predictions. I appreciated their GNN-based approach to this problem and feel that they have laid the groundwork for new and compelling approaches to tackle a complex problem. Fig. 5, which is much simpler than the others, is also much more interpretable.

The authors discuss the code review at length, but I also note that the code and data are now publicly available.

Minor issues:

Lines 173-174 refer to the models as having 8M and 150M params, respectively, while lines 192-193 refer to the model's 5M and 178M parameters. It is unclear if the earlier models were fine-tuned to create the latter models.

(Remarks on code availability)

Reviewer #2

(Remarks to the Author)

The paper introduces the Protein Set Transformer (PST), a protein-based genome language model that uses an encoder-decoder architecture to represent genomes as sets of proteins. The model, trained on over 100,000 viral genomes, demonstrates improved performance over existing homology- and language model-based approaches in representing viral genome relationships and identifying operationally related proteins. Key results include PST's ability to better cluster capsid proteins and other functionally related proteins, suggesting its utility in understanding viral genomics, ecology, and evolution.

The authors have presented a novel method for representing viral genomes that appears valid, particularly given the challenging problem of high genetic diversity and the lack of universal genes across viral species. The use of triplet loss for embedding training, along with data augmentation through PointSwap, contributes robustness to the model. However, the performance validation heavily relies on internal metrics without sufficient comparison against state-of-the-art techniques on a diverse and external dataset, limiting the evaluation's robustness. More details about how the model generalizes to different viromics tasks would further strengthen the validity. In particular, external validation using datasets such as NCBI's RefSeq viral database or the Global Virome Project's dataset would provide more comprehensive insights.

The validation dataset could be expanded to include more diverse viral genomes, particularly those from underrepresented environments such as soil and freshwater ecosystems. This could be achieved by incorporating data from the Earth Microbiome Project and Tara Oceans expeditions, which contain a wealth of under-sampled viral genomes. Such datasets would help assess PST's capability to generalize to a broader range of environmental conditions and host associations. The paper could be strengthened by including case studies that apply PST to specific viromics challenges. For instance, demonstrating PST's ability to predict host-virus relationships in marine and terrestrial environments using the Tara Oceans dataset or the EcoHAB project data would provide real-world examples of its applicability. Additionally, showcasing how PST can improve protein function prediction in poorly annotated viral families, such as those in the NCBI Viral Genomes Resource, would emphasize its practical impact.

(Remarks on code availability)

The code seems to work well. I could not verify every one of the figures generated, but the workflow seemed to work well.

Reviewer #3

(Remarks to the Author)

The authors present a Protein Set Transformer (PST) architecture for capturing viral genome relationships based on protein content. While the experiments and applications show the potential usage of PST, there are several major issues with the methodology and experiments. Addressing them would strengthen the manuscript and highlight the authors' contributions.

1. The idea and structure of the PST is not novel. There already exist several protein-based genome Transformer models [1, 2]. The author should mention and briefly discuss them in the introduction.

[1] <https://www.nature.com/articles/s41467-024-46947-9>

[2] <https://www.biorxiv.org/content/10.1101/2024.07.03.602011v1>

2. Leave-one-group-out cross validation is a very good idea to evaluate the robustness of the model and has been adopted by multiple publications. However, the author claims they always keep Duplodnaviria in the dataset is not reasonable. Duplodnaviria contains the majority of the protein diversity of the viruses and taxa within them share less homologous genes. Thus, if the authors want to keep as many data as possible and gain a more comprehensive results at the same

time, they should consider both 'class' and 'order' as group rather than realm.

3. The triplet loss training objective guided by genome-genome relatedness is an interesting approach. However, the goals and motivations of the loss is not entirely clear. The main idea behind triplet loss is to learn an embedding space where related examples (the anchor and positive) are pulled closer together, while unrelated examples (the anchor and negative) are pushed farther apart. However, the authors choose the next most related genome to the anchor after the positive as the negative. This is strange because viral species within each genus are usually highly similar and usually have the similar biological properties (close in phylogenetic tree, infecting same host, and share protein organization). But the PST model aims to separate them. Some studies/experiments are needed to show the effectiveness of this strategy.

4. The ablation study comparing PST to the protein language model (pLM) and nucleotide-based methods is insufficient to demonstrate the advantages of the triplet loss training objective. Both the pLM and nucleotide methods (GenSLM, HyenaDNA) were trained using masked language modeling (MLM) strategies, where the target tokens are amino acids or k-mers/nucleotides, respectively. To properly showcase the benefits of triplet loss over MLM, the authors should include an additional baseline where the PST architecture is trained via MLM instead. In this setup, the target tokens would be the individual protein embeddings within each genome, rather than the amino acid sequences. Comparing the genome clustering performance of this MLM-trained PST to the triplet loss-trained PST would provide a more direct assessment of the impact of the training objective, while controlling for the architectural differences between PST and the other baselines. Currently, it is unclear whether the improved clustering results are primarily due to the PST architecture better capturing higher-order protein organization, or if the contrastive triplet loss itself plays a key role by enforcing a more globular embedding space geometry. An MLM-trained PST baseline would help decouple these two factors.

5. It is not surprising that PST clusters viral genomes by taxonomic realm better than other methods in the first experiment. This is because the authors use triplet loss to constrain the model to output embeddings where similar genomes have close distances in the latent space. The experiment is unfair and cannot demonstrate that PST outperforms other methods because this loss function directly optimizes for the desired clustering outcome. To convincingly show the superiority of the PST architecture itself, the authors should consider training PST with a different self-supervised objective that does not explicitly optimize for genome-genome distances. This would reveal whether the protein set transformer model confers advantages in capturing taxonomic relationships, independent of the triplet loss.

6. The manuscript lacks important details about the computational resources and time required to run the PST model. Given the reliance on the large ESM2 language model, PST likely has substantial computational overhead compared to standard methods. To assess PST's practicality and scalability, the authors should report hardware specifications, runtimes, and memory consumption for PST in each experiment. Also, the author should provide a guideline of running it on real-world sized datasets (e.g. the running time per 10,000 viral genomes).

(Remarks on code availability)
See my comments above.

Version 1:

Reviewer comments:

Reviewer #2

(Remarks to the Author)
The authors have covered all of the comments and addressed them satisfactorily. I would accept the manuscript as is.

(Remarks on code availability)
The code works as described and I can generate most of the figures in the manuscript.

Reviewer #3

(Remarks to the Author)
The authors have properly addressed my comments. I only have one remaining question.

An issue that could inflate performance of PST tools is the data leakage from pretraining data – it has been shown that models achieve higher values of the performance metrics on species, DNA regions and protein families that were included in the pretraining data of the foundation models (<https://www.biorxiv.org/content/10.1101/2024.03.07.584001v1>, <https://proceedings.mlr.press/v261/hermann24a.html>, <https://www.biorxiv.org/content/10.1101/2025.01.22.634321v1>). This is because even though labels for those DNA/protein regions might not be available at the supervised fine-tuning stage, such sequences are not really novel (as in: unseen by the model at any stage of the training). Pre-training data leakage might artificially inflate the reported performance metrics; a problem that the compared methods might not be affected by. Ideally, the authors should check the performance of PST on sequences that were not included in the pretraining sets of ESM or retain the ESM on the same training data.

(Remarks on code availability)

Reviewer #4

(Remarks to the Author)

I am writing as a substitute for Reviewer #1. I have carefully reviewed the original comments alongside the authors' detailed response and revisions, and I am pleased to confirm that all of the initial concerns have been thoroughly addressed.

To further enhance the practical utility of this excellent work, I would like to offer one additional suggestion for the authors' consideration.

A potential barrier to the widespread adoption of the PST model could be its computational requirements. To help other researchers assess the feasibility of using this model in their labs, it would be helpful to expand on the resource requirements in the Results or Discussion section. While the current mention of training and inference times in the Method section is a good start, further details would be invaluable. For example, what is the minimum GPU memory required for the model to run effectively? How does the model perform on a CPU-only system? This information is particularly crucial for wet-lab-focused groups who may have limited access to specialized hardware like the A100.

(Remarks on code availability)

Dear Reviewers,

Thank you for your insightful comments on our work. We are glad the reviewers found our manuscript well-written, and our results clearly presented. We have made significant revisions including incorporating several new analyses to address the reviewers' concerns. All line numbers mentioned here refer to the clean manuscript.

On behalf of all authors,

Sincerely,

Cody Martin, first author

Karthik Anantharaman, corresponding author

SUMMARY OF MAJOR CHANGES

We highlight what we perceived to be the major requests from each reviewer and the actions we subsequently took to address these concerns.

1. Simplify the figures

- The primary contributor to figure complexity was considering various configurations for number of nearest neighbors and the clustering resolution for each analysis. We have thus simplified all figures by choosing a single set of clustering parameters (k=15, resolution=0.9) to use for all analyses as appropriate.
 - We notably also standardized all clustering and embedding similarity calculations to be done using angular similarity on unit-normalized embeddings. This led to the most consistent results among each set of evaluations.
- This was further complicated for protein embedding analyses, since we originally clustered proteins only within genome clusters. This led to a large number of genome embedding-protein embeddings combinations. We chose to cluster proteins globally, instead of per genome cluster, to reduce the combinatorial load. This also generally led to increased performance at the expense of somewhat greater computational processing for a large number of proteins.

2. Evaluate on different viral datasets that are less represented in the original test dataset

- We curated a 2nd test dataset solely comprised of viruses detected from soil metagenomes in MGnify to address the fact that our original test dataset from IMG/VR v4 was roughly 50% aquatic viruses. This was an important consideration since soil microbiomes are considerably more complex¹, so results extracted from aquatic ecosystems may not be representative of generalization to soil viruses. All PST evaluations were performed for both test datasets.

3. Compare the choice of training objective to decouple objective from model architecture

- We trained encoder-only PST models with a masked language modeling-style objective (lines 989–1018 for description of PST-MLM architecture and training). Our original encoder-decoder PST trained with triplet loss was superior across most evaluation tasks.

4. Compare cross validation strategies

- We implemented a 2nd CV strategy also using Leave-One-Group-Out with groups we built to have minimal overlap in protein diversity (Supplementary Fig. 26C, lines 1048–1083 for methodological description of how this was achieved).
- We compared models tuned using our original taxonomic approach with this protein diversity approach and did not generally find substantial differences between the PST models tuned with either approach, except for host prediction, which could be explained by differences in model size.

Other major changes not directly related to any of the above critiques:

1. **CHANGE:** All embeddings were searched/clustered using the exact same settings (L2 normalization, angular similarity, k=15, resolution = 0.9). This notably led to subtle changes throughout the manuscript, since most analyses relied on embedding searches/clustering.
 - a. **REASON:** This helped lead to more consistent results across all models and prevented potential bias when using Euclidean distance, since PST-TL models are optimized in Euclidean space.
2. **CHANGE:** Instead of computing a cluster-average amino acid identity (AAI) for all clusters produced by each genome embedding, we plotted AAI and average structural identity (ASI) for pairs of genomes against the embedding similarity for the corresponding pair. We then summarized this using Pearson correlation.
 - a. **REASON:** This helped simplify both our interpretation as well as the presentation of this result.
3. **CHANGE:** We removed the protein UMAPs from Figure 3.
 - a. **REASON:** This was changed since we were no longer clustering proteins within each genome cluster. This made it challenging to identify an example

case to display. However, the remaining results still demonstrate the utility of PST for protein tasks.

4. **CHANGE:** In Figure 3C–F and Supp. Figure 21, we do not show the quantification for detection of replication or packaging modules
 - a. **REASON:** These weren't detected in the co-clustering graphs, so we did not pursue them further.
5. **CHANGE:** In Figure 4E and Supp. Figure 22, we increased the number of neighbors considered for annotation transfer. Additionally, we tested an additional constraint to assess the confidence of annotation transfer.
 - a. **REASON:** This scenario is more realistic to how a protein function annotation tool might be trained using embedding similarity-based annotation transfer. Further, this simplified the original presentation of the data where we compared the slopes of these lines.

POINT BY POINT RESPONSES TO REVIEWER COMMENTS

Reviewer #1 (Remarks to the Author):

PROTEIN SET TRANSFORMER: A PROTEIN-BASED GENOME LANGUAGE MODEL TO POWER HIGH-DIVERSITY VIROMICS.

Cody Martin, Anthony Gitter, and Karthik Anantharaman

The paper by Martin, Gitter, and Anantharaman describes the creation of viral-specific protein language models and their use in various challenges holding back viral bioinformatics, including gene identification, protein annotation, and viral host prediction. Overall, the paper is well-written, grammatically well-formed, excellent and a novel contribution to the field, and I support its publication.

We thank the reviewer for their positive comment.

The pLMs described in this paper start with the ESM2 framework, and they have layered on additional, genome-specific attributes, notably the strand and location of the the genes that encode each protein. The novelty is enhanced in training the language models by introducing triplet training to provide both positive and negative mining of closely related genomes. This approach allows the authors to establish point swapping in the architecture to simulate evolution during data augmentation and then use the augmented data in the loss function.

The authors tuned two models—a small model with six layers and 8 million parameters and a larger model with 30 layers and 150 million parameters, and the training took proportionally longer for the large model, as expected.

They applied the learned Protein Set Transformer models to a testing set of viral genomes and demonstrated that the model learns “meaningful” information about the genome. Figure 2 demonstrates that new information has been learned and that the protein methods generate more meaningful information than the DNA-based methods.

I was excited to see that the results here recapitulate other results demonstrating that protein-based methods are far superior to nucleotide methods when clustering viral genomes. I also appreciated the limitations the authors highlighted in lines 256-261, comparing their results to existing taxonomy and host predictions.

I found several figures, especially Figs. 2-4, to be overly complex and confusing because they contain too much information, much of which is unnecessary. For example, there is no description in the Fig 2 legend about the clustering resolution, and I am not sure it adds much to the story because it is yet another parameter that can be varied.

We thank the reviewer for this comment. We agree that the original figures were too complex (and also resulted from overcomplicated evaluations!). We chose to redo all analyses using a single set of clustering hyperparameters ($k=15$, resolution=0.9) to simplify the number of subpanels in each figure. Additionally, we no longer clustered protein embeddings per genome cluster, instead clustering proteins globally across each dataset, for 2 reasons: (1) This actually improved performance for most methods, and (2) This reduced an issue considering different combinations of genome and protein embeddings for clustering.

As a specific example, **Fig. 2BC** now only show 4 bar plots instead comparing structure-based genome-genome relatedness with embedding similarity instead of the 10 subplots previously.

Line 250 says, “Notably, pst-small genome clusters have the highest AAI among all methods (Fig. 2E)”, but it also has more, smaller clusters, so just the highest AAI is not that notable.

We thank the reviewer for this comment and note that this sentence has been removed due to changes we made in performing this analysis in comparing genome embeddings with external measures of genome-genome relatedness like AAI.

Next, the authors tackled the thorny subject of viral protein annotations, and their data in Fig. 3 demonstrates well-defined protein clustering with related functions.

They have clearly demonstrated the benefits they have brought to the data by showing that ESM2 embeddings did not lead to interpretable functional modules (lines 349-351), whereas their vPST, including genome location, adds functionality. Again, I think Fig. 3 could be simplified for clarity. For example, Fig 3 D PHROG panels show no difference, and perhaps Fig 3C could be simplified to fewer examples. I am not sure the figure will be improved, but the scaled attention in Fig. 3A should be the same scale for PHROG or VOG. Alternatively, you might consider eliminating PHROG or VOG data from the entire figure.

We thank the reviewer for this comment. This figure changed overall due to the changes in clustering protein embeddings, which caused differences to become more noticeable throughout the panels. To address the comment about Fig. 3C (which is now **Fig. 3B**), we only included 3 examples instead of 5 for the co-clustering summary networks. **Fig. 3A** now is plotted on a relative scale and only shows PHROG categories (with VOG shown in **Supplementary Fig. 14**) since the pattern was somewhat clearer.

Fig. 4 could also be simplified by removing the genome clustering resolution – especially as that is secondary to the story. In panel C, the authors used low and med resolution, but in panel D, they used med and high resolution, which provides the perception of cherry-picked data.

We thank the reviewer for this comment. In the original version, the resolutions in panel C and D did not refer to the same thing. However, we have removed all mention of clustering resolution since we used the same clustering settings for all embeddings. This led to a simplified **Fig 4**.

Finally, the authors tackle the challenge of host-species predictions. I appreciated their GNN-based approach to this problem and feel that they have laid the groundwork for new and compelling approaches to tackle a complex problem. Fig. 5, which is much simpler than the others, is also much more interpretable.

We thank the reviewer for this comment.

The authors discuss the code review at length, but I also note that the code and data are now publicly available.

We included this extensive discussion because we are sensitive to public concerns about dual-use research pertaining to machine learning and virology. An ongoing study by Epoch AI (<https://sentinelbio.org/grant/epoch-ai/>) that tracks leading biological machine learning models, including ours, reports that only a tiny fraction discusses dual-use or biosecurity considerations. We have found that policy and government groups valued this discussion. However, we have revised it to account for the code review, code and data release, and recent computational biosecurity literature.

Minor issues:

Lines 173-174 refer to the models as having 8M and 150M params, respectively, while lines 192-193 refer to the model's 5M and 178M parameters. It is unclear if the earlier models were fine-tuned to create the latter models.

We thank the reviewer for this comment. The first line is now discussed in **line 220**, which we have modified to:

“We tuned eight different PSTs **starting** with small (6-layer, 8M param) or large (30-layer, 150M param) ESM2 protein embeddings, respectively...”

to indicate that the model parameter sizes here refer to the ESM2 pLMs used as inputs for PST. The size for the actual PST models we trained are described in **lines 261–263** and reported in **Supplementary Table 11**.

Reviewer #2 (Remarks to the Author):

The paper introduces the Protein Set Transformer (PST), a protein-based genome language model that uses an encoder-decoder architecture to represent genomes as sets of proteins. The model, trained on over 100,000 viral genomes, demonstrates improved performance over existing homology- and language model-based approaches in representing viral genome relationships and identifying operationally related proteins. Key results include PST's ability to better cluster capsid proteins and other functionally related proteins, suggesting its utility in understanding viral genomics, ecology, and evolution.

The authors have presented a novel method for representing viral genomes that appears valid, particularly given the challenging problem of high genetic diversity and the lack of universal genes across viral species. The use of triplet loss for embedding training, along with data augmentation through PointSwap, contributes robustness to the model. However, the performance validation heavily relies on internal metrics without sufficient comparison against state-of-the-art techniques on a diverse and external dataset, limiting the evaluation's robustness.

We thank the reviewer for this comment, but we want to emphasize that the evaluations in the study were extremely challenging since genome language models (**especially** protein-based genome language models) are very new to biological AI modeling. This means that there are not really gold-standard benchmarks for evaluating performance of genome AI models, **especially for viral genomes**. For example, while Evo², a nucleotide genome language model, has evaluations with several prokaryotic datasets like protein fitness, such datasets do not exist for large, diverse sets of viruses. Further, we questioned what utility there would be in evaluating PST's predictive capabilities on limited subsets of viruses where these other types of data might be available when these smaller subsets of viruses may not be representative of all/a large fraction of viruses.

This is why we opted to compare PST embeddings with metrics that could be determined/computed for most viruses in our test datasets. However, even though these metrics are derived, they are still commonly used and interpretable, such as AAI. These metrics are also independent from PST embedding similarity.

Further, we actually did evaluate PST on two diverse test datasets that were completely distinct from our training dataset. One of these test datasets was 1.5x larger than our training dataset.

More details about how the model generalizes to different viromics tasks would further strengthen the validity.

We thank the reviewer for this comment. We want to point out that, although we only directly tested 1 viromics task (host prediction **Fig. 5**), the other results do suggest how PST would perform on other viromics tasks. For example, **Fig. 2** shows how PST can detect relationships even among very distantly related genomes, which could help to

improve viral taxonomy. Likewise, **Fig. 3 & 4** suggest an awareness of protein function, which could help with protein annotation.

We additionally want to emphasize that thorough studies demonstrating PST's applicability to downstream viromics tasks is beyond the scope of this study, since these analyses would not be trivial. In other words, this study is meant to serve as an initial platform for that future work. For example, we are currently planning detailed follow-up studies to apply PST to different viromics tasks, such as virus identification, virus-host prediction, and viral protein annotation.

In particular, external validation using datasets such as NCBI's RefSeq viral database or the Global Virome Project's dataset would provide more comprehensive insights.

We thank the reviewer for this comment, but we disagree that using these datasets would provide more comprehensive insights. First, not only is the Global Virome Project not publicly available, but it also seems to be heavily focused on a narrow subset of viruses with zoonotic potential that do not necessarily represent the vast diversity of viruses on the planet. Second, NCBI RefSeq does not include any additional information beyond the viral genomes and sometimes ORFs. This means that almost the exact same level of information is present in NCBI RefSeq compared to the training and test datasets used in this study. Further, 71.5% of our training dataset comes from IMG/VR v3, which already contains both NCBI RefSeq and NCBI GenBank viruses.

The only major difference is that NCBI RefSeq viruses are cultured by definition, so the hosts are known. We leveraged this for our host prediction proof-of-concept in **Fig. 5**, which exclusively used NCBI RefSeq viruses for training and NCBI GenBank viruses for evaluation.

The validation dataset could be expanded to include more diverse viral genomes, particularly those from underrepresented environments such as soil and freshwater ecosystems. This could be achieved by incorporating data from the Earth Microbiome Project and Tara Oceans expeditions, which contain a wealth of under-sampled viral genomes. Such datasets would help assess PST's capability to generalize to a broader range of environmental conditions and host associations. The paper could be strengthened by including case studies that apply PST to specific viromics challenges.

We thank the reviewer for this comment. The Earth Microbiome Project is not a centralized resource, which made it challenging to try to determine the feasibility of incorporating this resource. Additionally, the vast majority of ecosystems represented in the Earth Microbiome Project were either animal gut or aquatic systems, which are both well-represented in our datasets (**Supplementary Fig. 2C**). Further, we did include data from Tara Oceans (**Supplementary Fig. 2B**), as well as many other marine-based expeditions. However, we included viruses exclusively detected from soil metagenomes in MGnify as a 2nd test dataset for all analyses, since soil viruses only accounted for ~15% of our 1st test dataset but tend to be extremely diverse and complex. Nearly all evaluations suggested little difference in performance on either test dataset, except for some that

focused on protein function annotations. However, the difference in performance on this dataset in those cases could either be due to limited annotation capacity coupled with a reduced number of proteins OR due to intrinsic complexity and diversity of soil ecosystems.

As far as applying PST to specific viromics tasks, please see our response to the previous comments. We reiterate that **Fig. 5** is directly related to demonstrating PST for a host prediction viromics task, but all other figures also hint at the applicability of PST to downstream viromics tools.

For instance, demonstrating PST's ability to predict host-virus relationships in marine and terrestrial environments using the Tara Oceans dataset or the EcoHAB project data would provide real-world examples of its applicability.

We thank the reviewer for this comment. We heavily debated on whether this would be useful and ultimately decided that this would only complicate and not strengthen this study for several reasons.

One, as we have noted in the discussion (**lines 605–614** and **622–634**), there are numerous modifications we could have made to our proof-of-concept host prediction framework. In addition, the training datasets used by many other host prediction tools are very limited in diversity, so it is not clear how well they generalize to highly diverse sets of viruses (**Supplementary Fig. 24B** for iPHoP training viruses compared to the PST training and test datasets in **Supplementary Fig. 3**). Thus, a good host prediction tool will need to leverage much more representative datasets to achieve greater performance. These two changes alone constitute an out-of-scope concern for this paper; however, we note that we are actively pursuing this in a future study.

Second, we questioned what would be gained simply from predicting hosts in another dataset. In the most trivial setups, PST could predict a host for *any* virus if a sufficient number of hosts are included in the interaction graphs and if the confidence thresholds are lowered. However, simply predicting a host for a virus does not necessarily indicate anything of value without further benchmarks to externally validate if the prediction is reasonable. For example, the confidence scores are model derived and *not* based on some external assessment, such as considering if the virus and host could be found in the same ecosystem, sample, etc. This was possible for our host prediction application, since the viruses we evaluated had **known hosts**. This meant that we did not need to further assess the validity of host predictions here, just that we needed to ensure that the true host was predicted above a model-derived confidence metric. Taken together with the previous point, this further indicates the need for a separate thorough analysis to more sufficiently demonstrate PST's applicability to host prediction that is beyond the scope of this study.

Additionally, showcasing how PST can improve protein function prediction in poorly annotated viral families, such as those in the NCBI Viral Genomes Resource, would emphasize its practical impact.

We thank the reviewer for this comment. We now include an assessment of ability to improve functional annotation that was greatly expanded after revision (**Fig. 4E, Supplementary Fig. 22**). This improvement was based on proteins that could not be detected by VOG r219 profiles (i.e. no relatives) or had matches to VOG profiles of unknown function (i.e. has relatives that do not have a known function), instead of the NCBI Viral Genomes Resource. We did not, however, specifically indicate which VOG profiles had the greatest improvement in annotation rates, but this seems to be too granular for this study. Again, we also note that our datasets do actually include NCBI viruses but also include several other much larger, more diverse datasets as well. To expand on this, we will follow up on this with a future study that is dedicated to improving protein annotation for viral protein families.

Reviewer #2 (Remarks on code availability):

The code seems to work well. I could not verify every one of the figures generated, but the workflow seemed to work well.

Reviewer #3 (Remarks to the Author):

The authors present a Protein Set Transformer (PST) architecture for capturing viral genome relationships based on protein content. While the experiments and applications show the potential usage of PST, there are several major issues with the methodology and experiments. Addressing them would strengthen the manuscript and highlight the authors' contributions.

1. The idea and structure of the PST is not novel. There already exist several protein-based genome Transformer models [1, 2]. The author should mention and briefly discuss them in the introduction.

[1] <https://www.nature.com/articles/s41467-024-46947-9>

[2] <https://www.biorxiv.org/content/10.1101/2024.07.03.602011v1>

We thank the reviewer for this comment. We note that Hwang *et al.* (1st paper) was mentioned in the introduction (**lines 64–70**) and further compared against when describing our reasoning for a triplet loss objective in **lines 143–152**. Additionally, we trained a masked language modeling variant of PST that was similar to Hwang *et al.*, so we added discussion that refers to this direct comparison (**lines 188–192, 586–589**).

We have added **lines 70–73** to discuss the 2nd paper in the introduction that we are copying here for quick reference:

Further, similar work used similar protein-based genome modeling to classify plasmid proteins ontologically. Again, the supervised training of that study causes a lack of generality when considering that other related tasks could benefit from the same contextualization approach.

We also added a sentence in **lines 80–82** to explicitly state that PST has both algorithmic advances as well as is trained in a more general self-supervised setup (**lines 82–83**):

We developed several novel algorithmic advances for the PST architecture and implemented more memory-efficient data handling compared to similar methods. We additionally pretrained foundation PST models on >100k high-quality dereplicated viral genomes encoding >6M proteins using a self-supervised objective.

2. Leave-one-group-out cross validation is a very good idea to evaluate the robustness of the model and has been adopted by multiple publications. However, the author claims they always keep Duplodnaviria in the dataset is not reasonable. Duplodnaviria contains the majority of the protein diversity of the viruses and taxa within them share less homologous genes. Thus, if the authors want to keep as many data as possible and gain a more comprehensive results at the same time, they should consider both 'class' and 'order' as group rather than realm.

We thank the reviewer for this comment and agree that taxonomy is possibly not the best way to create broad groups of viruses for cross validation (CV). One reason that the reviewer has alluded to is that viral taxa, especially at the broader ranks like realm, are not homogeneous, and there is likely substantial diversity even within Duplodnaviria.

We chose not to focus on viral taxonomy for CV and **instead considered an alternate approach to address the reviewer's root concern about the protein diversity of these viruses**. We divided each training virus into 4 groups with minimally overlapping protein diversity (see **lines 1048–1083** for the methods and **Supplementary Fig. 26** for the confirmation). Briefly, we clustered genomes based on similar protein content profiles using the dice score. Then, we seeded each of the 4 protein diversity CV groups with the 4 largest clusters that did not overlap significantly in protein diversity (**Supplementary Fig. 26B**). Then we used a greedy algorithm to expand each of these CV groups based on the average dice score between all pairs of genomes from each remaining cluster to the seeding clusters.

We kept the models tuned with taxonomy ("-T" models) to compare with the models tuned with protein diversity ("-P" models). There were not generally major differences in performance between these 2 CV strategies, except for host prediction (which could also be explained by other factors). Further, since each test dataset was primarily composed of Duplodnaviria (**Supplementary Fig. 2C**) that were distinct at the protein level from the training viruses (**Supplementary Fig. 3BD**), this suggests that the taxonomy models could still be valuable in the future.

3. The triplet loss training objective guided by genome-genome relatedness is an interesting approach. However, the goals and motivations of the loss it not entirely clear. The main idea behind triplet loss is to learn an embedding space where related examples (the anchor and positive) are pulled closer together, while unrelated examples (the anchor and negative) are pushed farther apart. However, the authors choose the next most related genome to the anchor after the positive as the negative. This is strange because viral species within each genus are usually highly similar and usually have the similar biological properties (close in phylogenetic tree, infecting same host, and share protein organization). But the PST model aims to separate them. Some studies/experiments are needed to show the effectiveness of this strategy.

We thank the reviewer for this comment and will clarify the utility of triplet loss here.

Our triplet sampling strategy is based on a study with deep learning for pointsets that uses triplet loss³. However, there are other examples that specifically discuss triplet sampling strategies^{4,5} and supported our triplet sampling strategy.

In cases with target labels, hard negative sampling could be performed by choosing the absolute closest example that belongs to a different class. However, PST is trained with a self-supervised objective, so we do not have any positive/negative labels. Thus, we use semi-hard negative sampling in which the negative example is chosen to be the next closest genome in PST embedding space that is farther than the positive example from

the anchor. It is important to choose “hard” or “semi-hard” negative samples since this will lead to faster updates while training the model. Choosing the absolute farthest negative example from the anchor (i.e. an easy negative) would lead to insignificant contributions to the loss as these “easy” examples already satisfy the triplet loss constraints, meaning that these would result in slow convergence. Meanwhile, randomly choosing a non-positive genome as the negative would lead to inconsistent contributions to the loss due to sometimes choosing “hard” cases and sometimes choosing “easy” cases.

Since we trained PST in a self-supervised manner, we do not know whether the choice of positive or negative genomes are actually good. Thus, our implementation includes a weighting factor based on the choice of negative examples (see **lines 929–941, 965–971**). This weight is calculated using the Chamfer distance of the input ESM2 protein embeddings for all pairs of genomes in a minibatch (see **Equation 2 and 3**). This weight is a negative exponential decay factor, meaning that it ranges from [0, 1]. For example, if the anchor and negative genomes are actually highly related, then the Chamfer distance will be near 0, and subsequently the weight factor will approach 1. In this case, holding the anchor-positive distance constant, the contribution of this triplet will be relatively small to the loss since the weighted anchor-negative distance is subtracted from the anchor-positive distance (**line 909**).

With the same anchor and positive genome, if the choice of negative is actually good since the negative is not related to the anchor, then the Chamfer distance will be large, and the weight factor will approach 0. In this case, considering the same choice of positive genome, the loss will be greater, resulting in a non-0 contribution to the loss that will lead to model parameters updates after backpropagation and gradient descent. **Thus, the negative exponential decay scale factor for the choice of negative example modulates how reasonable of a choice the negative example is during loss calculation.**

This is further corrected by the use of a minimal distance margin (**Fig. 1D**). This margin α is how much closer the positive example needs to be to the anchor compared to the negative before this triplet stops contributing to the loss entirely. This should help to prevent micro-optimizations that could lead to trivial solutions like embedding the anchor and positive genomes in the exact same location. **More plainly stated, this margin α prevents repelling the negative genome an infinite amount of distance away from the anchor.**

Finally, while the reviewer suggests that triplet loss may not be beneficial due to our semi-hard negative sampling strategy since viruses of the same genus are highly related, our above description indicates why this is not a concern due to the weighting factor used in the loss function. However, we wanted to further address this. One, since we have dereplicated our datasets (**Supplementary Fig. 3**), it is not as likely that there are numerous examples of these extremely closely related viruses in our datasets. Second, both our positive and negative sampling happen online (i.e. not fixed). This means that even if a poor choice of negative is chosen (i.e. highly related to the anchor), this does not guarantee that the same poor choice is made for the next minibatch or even the next

epoch. Third, practically speaking, we believe that a model capable of distinguishing closely related examples should also be capable of distinguishing more distantly related/unrelated examples. Since this is a bit speculative, we have also trained a PST variant trained with masked language modeling instead of triplet loss as per this reviewer's following recommendations. The results shown throughout the manuscript nearly unanimously agree on the superiority of triplet loss for most of the evaluations in this study. Most notably, Fig. 2 shows that PST models trained with triplet loss more accurately relate genomes than the PST trained with MLM.

4. The ablation study comparing PST to the protein language model (pLM) and nucleotide-based methods is insufficient to demonstrate the advantages of the triplet loss training objective. Both the pLM and nucleotide methods (GenSLM, HyenaDNA) were trained using masked language modeling (MLM) strategies, where the target tokens are amino acids or k-mers/nucleotides, respectively. To properly showcase the benefits of triplet loss over MLM, the authors should include an additional baseline where the PST architecture is trained via MLM instead. In this setup, the target tokens would be the individual protein embeddings within each genome, rather than the amino acid sequences. Comparing the genome clustering performance of this MLM-trained PST to the triplet loss-trained PST would provide a more direct assessment of the impact of the training objective, while controlling for the architectural differences between PST and the other baselines. Currently, it is unclear whether the improved clustering results are primarily due to the PST architecture better capturing higher-order protein organization, or if the contrastive triplet loss itself plays a key role by enforcing a more globular embedding space geometry. An MLM-trained PST baseline would help decouple these two factors.

We thank the reviewer for this comment. We trained an encoder-only PST with a MLM objective using the same CV strategies (see lines 989–1018 for our specific implementation). Nearly all results show that PST-TL models were superior to the PST-MLM models, regardless of evaluating performance of genome (Fig. 2BC, 5B) or protein embeddings (Fig. 3B–F, 4CD).

One explanation for this overall superior performance of PST-TL is that, other than the optimal hyperparameters, the only major difference between PST-TL and PST-MLM is that PST-MLM does not directly output genome embeddings. Thus, PST-MLM genome embeddings were produced by mean pooling the contextualized protein embeddings, which is typical for creating “sentence”-level embeddings from “word”-level embeddings. PST-TL models, however, have a trainable decoder that learns to optimally pool contextualized protein embeddings to satisfy the triplet loss objective. This trainable decoder of the PST-TL could better help with genome-level evaluations compared to PST-MLM. However, it is still notable that PST-TL outperforms PST-MLM even on protein-level tasks.

5. It is not surprising that PST clusters viral genomes by taxonomic realm better than other methods in the first experiment. This is because the authors use triplet loss to constrain the model to output embeddings where similar genomes have close distances in the latent

space. The experiment is unfair and cannot demonstrate that PST outperforms other methods because this loss function directly optimizes for the desired clustering outcome. To convincingly show the superiority of the PST architecture itself, the authors should consider training PST with a different self-supervised objective that does not explicitly optimize for genome-genome distances. This would reveal whether the protein set transformer model confers advantages in capturing taxonomic relationships, independent of the triplet loss.

We thank the reviewer for this comment. Please see our previous response that also addresses this comment.

6. The manuscript lacks important details about the computational resources and time required to run the PST model. Given the reliance on the large ESM2 language model, PST likely has substantial computational overhead compared to standard methods. To assess PST's practicality and scalability, the authors should report hardware specifications, runtimes, and memory consumption for PST in each experiment. Also, the author should provide a guideline of running it on real-world sized datasets (e.g. the running time per 10,000 viral genomes).

We thank the reviewer for this suggestion to add information about computational resources of PST.

We added a paragraph (**lines 789–798**) in the methods that indicates processing rates for generating ESM2 embeddings, which is by far the most intensive step for the entire PST workflow. On a realistic dataset of ~10k genomes encoding ~150k proteins (such as the MGnify test set), ESM__large embeddings (640-dimensional) would take only about 18 min using 1 A100 GPU to sequentially embed minibatches of protein sequences. We estimate this would take ~6h using 128 CPUs (AMD EPYC 7H12) or 96h with 8 CPUs. While this may seem significant for users without GPU access, the runtime of many other bioinformatics tools is not significantly different. Additionally, PST uses frozen ESM2 embeddings as inputs, so this only needs to be computed once and can be saved to train many different PSTs, which is what we did here. The storage requirements are fairly slim since we store these embeddings using HDF5 files, which stores tensors as bytes instead of as text and can also be compressed.

Supplementary Table 11 contains the time for training the final best model for each PST using 1 A100 GPU and ranged from 3.9–33.7h. The inference times are described in **lines 889–903** and are fairly trivial compared to the time it takes to get the initial ESM2 embeddings. For example, the realistic dataset above would take <1 min using 1 A100 GPU. Using 128 CPUs, the inference time is actually about the same since there are some CPU-only bottleneck steps when loading the data, but using fewer CPUs would of course lead to increases in inference time. However, an estimation of using 8 CPUs would suggest a runtime of ~11 min on the sample dataset described above. Thus, total inference times including ESM2 inference would range from ~19 min using 1 GPU or up to ~96h with only 8 CPUs. The memory usage of PST inference is negligible since we prioritized memory efficiency when developing the both the data loading and model

processing architecture (~1.1GB with ESM__small inputs or ~3.3GB with ESM__large inputs, which both include the memory to fully load each dataset into memory).

We did not track memory usage during our evaluations in this study since the only difference between PST and the other methods would have been from embedding search/clustering due to the different embedding sizes. All other memory usage would have been roughly equivalent after that step. We, however, used the faiss library for embedding similarity search, which can be tuned to adjust memory usage as needed to balance processing speed with memory usage. For example, using the search setup described in **lines 1155–1160**, the memory usage ranged from 1.3–2.4GB for the MGnify dataset.

Reviewer #3 (Remarks on code availability):

See my comments above.

We have added this information to the GitHub page.

References

1. Roux, S. & Emerson, J. B. Diversity in the soil virosphere: to infinity and beyond? *Trends in Microbiology* **30**, 1025–1035 (2022).
2. Nguyen, E. *et al.* Sequence modeling and design from molecular to genome scale with Evo. *Science* **386**, eado9336 (2024).
3. Arsomngern, P., Long, C., Suwajanakorn, S. & Nutanong, S. Towards Pointsets Representation Learning via Self-Supervised Learning and Set Augmentation. *IEEE Transactions on Pattern Analysis and Machine Intelligence* **45**, 1201–1216 (2023).
4. Schroff, F., Kalenichenko, D. & Philbin, J. FaceNet: A unified embedding for face recognition and clustering. in *2015 IEEE Conference on Computer Vision and Pattern Recognition (CVPR)* 815–823 (IEEE, 2015). doi:10.1109/CVPR.2015.7298682.
5. Xuan, H., Stylianou, A. & Pless, R. Improved Embeddings with Easy Positive Triplet Mining. in *2020 IEEE Winter Conference on Applications of Computer Vision (WACV)* 2463–2471 (IEEE, 2020). doi:10.1109/WACV45572.2020.9093432.

REVIEWERS' COMMENTS

Reviewer #2 (Remarks to the Author):

The authors have covered all of the comments and addressed them satisfactorily. I would accept the manuscript as is.

We appreciate the reviewer's feedback throughout this process.

Reviewer #2 (Remarks on code availability):

The code works as described and I can generate most of the figures in the manuscript.

Reviewer #3 (Remarks to the Author):

The authors have properly addressed my comments. I only have one remaining question.

An issue that could inflate performance of PST tools is the data leakage from pretraining data – it has been shown that models achieve higher values of the performance metrics on species, DNA regions and protein families that were included in the pretraining data of the [foundation](https://www.biorxiv.org/content/10.1101/2024.03.07.584001v1) [models](https://proceedings.mlr.press/v261/hermann24a.html) (<https://www.biorxiv.org/content/10.1101/2024.03.07.584001v1> <https://proceedings.mlr.press/v261/hermann24a.html> <https://www.biorxiv.org/content/10.1101/2025.01.22.634321v1>). This is because even though labels for those DNA/protein regions might not be available at the supervised fine-tuning stage, such sequences are not really novel (as in: unseen by the model at any stage of the training). Pre-training data leakage might artificially inflate the reported performance metrics; a problem that the compared methods might not be affected by. Ideally, the authors should check the performance of PST on sequences that were not included in the pretraining sets of ESM or retain the ESM on the same training data.

We appreciate this feedback from the reviewer. We agree that data leakage and differences in training data among all the models we tested could have impacted the results presented here, which also makes evaluating various models challenging.

Since we performed ablation studies of PST, we have direct comparisons of ESM2-only information vs PST-specific information. In most analyses, PST outperformed ESM2 significantly, suggesting that the ESM2 pretraining data did not play a major role in performance here. We acknowledge that this does not directly disentangle the ESM2 pretraining data from performance when comparing against the other models, however, these other models sometimes outperformed ESM2. One explanation for this is that GenSLM, for example, was exclusively trained on prokaryotes and viruses, while ESM2 was trained on UniRef, which primarily has non-viral protein sequences.

A further challenge with filtering UniRef is that the sequences are not easily available, and the only available source (that we know of) from mmseqs does not provide sequence

deposit dates, meaning that we cannot know what sequences were specifically used to train ESM2.

Reviewer #4 (Remarks to the Author):

I am writing as a substitute for Reviewer #1. I have carefully reviewed the original comments alongside the authors' detailed response and revisions, and I am pleased to confirm that all of the initial concerns have been thoroughly addressed.

To further enhance the practical utility of this excellent work, I would like to offer one additional suggestion for the authors' consideration.

A potential barrier to the widespread adoption of the PST model could be its computational requirements. To help other researchers assess the feasibility of using this model in their labs, it would be helpful to expand on the resource requirements in the Results or Discussion section. While the current mention of training and inference times in the Method section is a good start, further details would be invaluable. For example, what is the minimum GPU memory required for the model to run effectively? How does the model perform on a CPU-only system? This information is particularly crucial for wet-lab-focused groups who may have limited access to specialized hardware like the A100.

We appreciate the suggestion from the reviewer. The memory requirements were specified in the Methods (lines 763–765). We have also implemented lazy data loading as the default inference setting, so the primary memory usage is from loading the model fully, which only requires 3GB for the largest models. We have described this and other strategies for lowering the memory footprint during inference in the Methods (lines 767–771).

The light memory requirements mean that cheaper GPUs are still useable, such as might be freely available using cloud GPUs from a Google Colab notebook, which we are actively trying to develop to broaden the usage. GPUs that are less powerful than A100/H100s will still result in orders of magnitude greater inference and could enable further training derived from PST.

Lines 757–760 and the GitHub describe the expected runtime for inference on CPUs (with all the other information also present in the GitHub) for a “typical” dataset that is more reflective, or even slightly larger, of what a small study would be focused on. Generating ESM2 embeddings is the longest step, especially without any GPUs, but the runtime is still comparable to other bioinformatics tools that would likely also be ran.